# Walking modulates visual detection performance according to stride cycle phase

**Matthew J. Davidson** [1] ✉, **Frans A. J. Verstraten** [1] & **David Alais**[1]

Walking is among our most frequent and natural of voluntary behaviours, yet the consequences of locomotion upon perceptual and cognitive function remain largely unknown. Recent work has highlighted that although walking feels smooth and continuous, critical phases exist within each step for the successful coordination of perceptual and motor function. Here, we test whether these phasic demands impact upon visual perception, by assessing performance in a visual detection task during natural unencumbered walking. We finely sample visual performance over the stride cycle as participants walk along a smooth linear path at a comfortable speed in a wireless virtual reality environment. At the group-level, accuracy, reaction times, and response likelihood show strong oscillations, modulating at approximately 2 cycles per stride (~2 Hz) with a marked phase of optimal performance aligned with the swing phase of each step. At the participant level, Bayesian inference of population prevalence reveals highly prevalent oscillations in visual detection performance that cluster in two idiosyncratic frequency ranges (2 or 4 cycles per stride), with a strong phase alignment across participants.

It is vital for our survival to correctly perceive and act within a dynamic environment. While the world is self-evidently dynamic, so too is the perceptual observer. In a typical day, we may make over 150,000 saccades, 15,000 head-turns and take several thousand steps[1–3]. To test perceptual processes in contexts that are closer to which evolution has optimised them, we need to take experiments out of darkened, static laboratories and into more complex, active situations. Here we examine the influence of the active observer during walking and test continuous visual performance, specifically focusing on the time-course of performance within the stride cycle. Despite the thousands of steps we take each day, the influence of locomotion upon perception is largely unexplored and within-gait modulations have received no attention. Previously, the technical limitations of controlling an environment for active perception have been high, but here we use wireless and position-tracked virtual reality to probe visual detection performance continuously during locomotion. We find that walking entrains rhythmic changes to perceptual performance within each stride cycle, despite the seemingly continuous and effortless nature of this everyday behaviour.

Studies examining the effects of walking and exercise on cognitive function have usually focused on changes i performance over an extended period of activity rather than effects occurring within the stride cycle. These studies have found that when various forms of light exercise and stationary conditions are compared, periods of moderate exercise can induce small increases in performance on cognitive tasks (reviewed in[4]) and in neuroplasticity[5]. Others have focused on how the presence of a dual-task during walking affects gait parameters such as speed and variability[6]. More recent reports have demonstrated that walking or light exercise can enhance visual processing relative to stationary conditions[7–12]. These results echo earlier work in animal models which has demonstrated how locomotion elevates the response gain of early visual processing areas—potentially providing a mechanism for enhanced performance on visual processing tasks[13–17]. Critically, however, most of these studies have focussed on average performance over the exercise period or immediately after exercise and have not studied temporal changes within the stride cycle.

There are a variety of findings that suggest visual performance might vary over the stride cycle. Recent investigations have demonstrated that responses to visual information depend on the phase of

[1]School of Psychology, The University of Sydney, Sydney, Australia. ✉e-mail: matthew.davidson@sydney.edu.au

human locomotion[18–21]. Humans capitalise on the pendulum-like momentum of each step when planning subsequent footsteps[22], and visual information about the upcoming terrain must be received within a critical window prior to heel strike for smooth locomotion to occur[21]. In addition, eye-movements[20,23–25] and the accuracy of visuomotor coordination[26] also show a coupling to the phase of locomotion. Together, there are a range of dynamic demands that occur with each step which may result in modulations of visual processing while we walk.

Here we introduce a method that allows us to finely sample performance in a simple visual detection task while walking to test for modulations over the time-course of the stride cycle. During natural unencumbered walking, we found that average accuracy, reaction times, and the likelihood of manual responses on a visual detection task all oscillated within the stride cycle in primarily two frequency bands (~2 and 4 Hz). These findings show that the continuous demands of walking impose rhythmic changes on both sensory processing and manual responding with every step. In doing so, we also validate our fine-sampling approach that will enable further investigation of stride cycle modulations of perception and performance.

## Results

We examined performance on a simple visual detection task during natural walking within a wireless virtual reality environment. Participants responded as quickly and accurately as possible to brief targets appearing within a circular annulus that drifted slowly and randomly around the screen (see example video and Fig. 1). Below, we first

compare average performance between walking and stationary conditions before turning to our main focus, which was whether performance would modulate relative to the phase of the stride cycle.

### Walking increases the threshold for visual target detection

Throughout the experiment, a continuous adaptive staircase procedure was implemented which manipulated target contrast to maintain participant accuracy at approximately 75%. Confirming this, average accuracy was close to 75% and not significantly different between stationary (M = 0.74, SD = 0.03) and walking conditions (M = 0.75, SD = 0.03, $t(35) = -0.99$, $p = 0.32$, $d = -0.16$, 95% CI [−0.01, 0.004], two-tailed). The stationary condition, however, was clearly easier for participants because the staircase settled at a lower average target contrast (M = −7.12 dB, SD = 0.24) than in the walking condition (M = −6.87, SD = 0.50; $t(35) = -4.07$, $p < 0.001$, $d = -0.70$, 95% CI [−4.08, −4.06], two-tailed).

While titrating contrast to maintain overall accuracy at 75%, target contrast on individual trials was presented at ±3 intensity levels centred on the estimate required for 75% performance to allow psychometric functions to be fitted over 7 contrast levels. Psychometric fits to participant accuracy revealed that the accuracy difference between standing and walking conditions was driven by a change in threshold ($t(35) = -3.50$, $p = 0.0013$, $d = 0.61$, 95% CI [−3.51, −3.49], two-tailed). There was no statistically significant change in the slope parameter between stationary and walking conditions ($t(35) = -1.92$, $p = 0.063$, $d = -0.45$, 95% CI [−0.03, 0.001], two-tailed). Interestingly, reaction times to targets were faster on average in walking compared

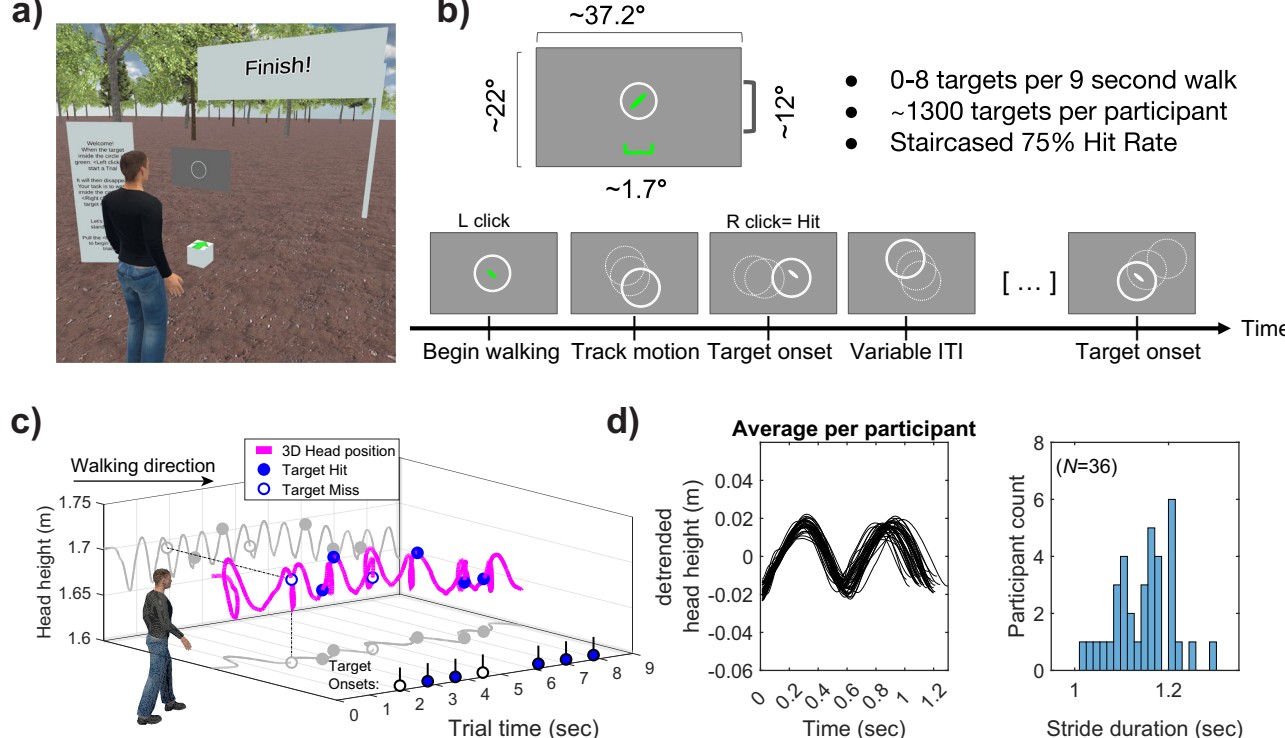

**Fig. 1 | Environment and trial structure. a** Third-person view of the virtual environment. Participants were positioned behind a virtual grey screen displaying the target stimulus. During the trial, the screen progressed with smooth linear locomotion at a constant velocity, in line with a small walking guide (three-dimensional animated game object). The avatar shown is for illustrative purposes only and was not present during the experiment. **b** The visual detection task required participants to monitor a drifting circular annulus. Small target ellipses (~1.7 d.v.a, 20 ms duration, illustration not to scale) appeared with a variable inter-trial interval (ITI), responses were provided via right trigger click. **c** Example data

from a single walking trial. The three-dimensional head position is recorded at 90 Hz (shown in magenta). Walking produces a stereotyped sinusoidal pattern of head motion on the vertical axis (head height, 2D projection on the back wall shown in grey). Peaks and troughs in head height correspond to the swing and stance phases of each step, respectively (see Methods). **d** (Left) Average detrended head height for each participant over their respective stride cycle. (Right) Distribution of average stride cycle duration across participants. Our primary interest was whether the timing of target onset relative to stride cycle phase would modulate task performance.

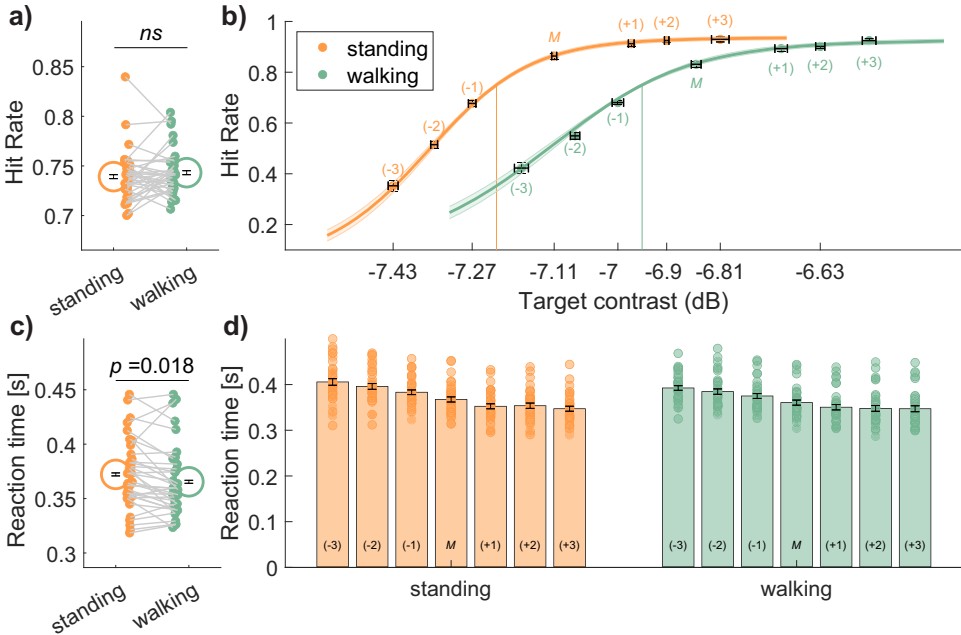

**Fig. 2 | Comparison of detection, hit rate and reaction time when standing and walking. a** On average, no statistically significant difference in accuracy between conditions as a result of the adaptive staircase procedure. Orange colours indicate standing, and green indicates walking. Grey lines link participant data; circles display the group mean, and error bars display ±1 SEM corrected for within-participant comparisons[86]. **b** Psychometric fits to hit-rate data for each condition. The staircase mean for each condition is indicated M, and contrast steps (see Methods) are indicated numerically in the range of −3 to +3. Vertical error bars represent ±1 SEM for hit rate, and horizontal error bars represent ±1 SEM for adaptive target contrast. **c** Reaction times for correct detection decreased when walking, compared to standing (colours as in (**a**)), and **d** reaction times decreased as accuracy improved over the psychometric function.

to stationary conditions ($t(35) = 2.47$, $p = 0.018$, $d = 0.41$, 95% CI [2.46, 2.48], two-tailed). There was also an inverse relationship between reaction times and accuracy across intensity levels, indicating participants were more accurate when responding quickly. A $2 \times 7$ repeated measures ANOVA (walking condition × contrast level) indicated main effects of condition ($F(1,35) = 6.12$, $p = 0.018$, $\eta_p^2 = 0.15$) and contrast (Mauchly's test of sphericity violated, Greenhouse–Geisser corrected, $F(2.70, 94.26) = 118.60$, $p < 0.001$, $\eta_p^2 = 0.77$) with no statistically significant interaction ($F(3.23, 113.10) = 2.07$, $p = 0.10$, $\eta_p^2 = 0.06$). Figure 2 displays a summary of this data.

## Performance oscillates within the stride cycle

We next compiled the hundreds of detection data points (M = 885.52, SD = 63.33 per participant) into a single stride cycle in order to test for temporal modulations of visual performance. For this analysis, we first epoched all walking trials into stride cycles (two sequential steps) by running a peak-detection algorithm on the head height time-series recorded on each trial. This analysis revealed that step and stride durations varied across participants (step duration M = 0.59 s, SD = 0.034; stride M = 1.18, SD = 0.069). Supplementary Fig. 4 visualises an example participant and the group-level stride cycle variability. As is common in gait-based research, we next resampled all strides to 1–100% stride completion to facilitate within- and across-participant averaging for our main effect of interest—whether the relative phase of an individual's stride cycle would modulate visual detection performance. Target onsets, which were presented at random times, could thus be allocated to a position relative to the stride within which it occurred (from 1 to 100%; see Fig. 3a–c for key stages of this workflow). In effect, the hundreds of data points are pooled into a single, densely sampled stride that can be resolved into fine time bins (e.g. 2.5% bin width in Fig. 3) and reliably analysed for temporal modulations. Figure 3c plots target density over the stride cycle and validates our random-probing and stride epoching procedure as the distribution is flat with ~18 (M = 17.84, SD = 1.2) targets falling in each bin.

The stride cycle analysis revealed clear oscillations in performance on the visual detection task. Figure 3d–f displays the group average results ($N = 36$) when binned into 40 linearly spaced, non-overlapping bins. The solid curves display the best-fitting first-order Fourier model (see Eq. 1). Figure 3g shows the goodness-of-fit ($R^2$) for all Fourier frequencies in the range from 0.2 to 10 cycles per stride (cps; in steps of 0.2 cps). For accuracy, the best-fit to performance over the stride cycle was at 1.93 cps ($R^2 = 0.48$), and group-level fits were above the 95th percentile of the $R^2$ distribution obtained by permutation ($n = 1000$) for the range 1.80–2.20 cps. For reaction times, the best-fitting oscillation was at 2.02 cps (and was significant over the range 1.60–2.40 cps), and for manual response likelihood, the best-fitting oscillation was at 1.90 cps (significant from 1.60 to 2.20 cps).

We performed several control analyses to confirm these oscillations were driven by the stride cycle. We first examined performance with respect to trial-onset, using clock-time (rather than stride cycle) as our reference for entrainment. For this analysis, we averaged each of the behavioural variables considered based on consecutive 1 s epochs in both the walking and standing data. Although these 1 s epochs are a rough approximation of average stride duration in our sample, the fit-strength of Fourier models was far weaker than that based on stride alignment (Supplementary Fig. 5). As expected, equivalent results were found in the same analysis applied to standing trials (Supplementary Fig. 6). These results provide complementary evidence that stride phase aligns the modulations in behavioural performance we report, rather than an alternative explanation based on trial-time.

## Population prevalence of stride cycle oscillations

In addition to the group-level test of oscillations in performance based on the stride cycle, we tested the population prevalence of stride cycle oscillations within our sample. Qualitative observations of our data showed some participant-level Fourier fits occurred outside the group-level best fit of approximately 2 cps. We therefore formally quantified

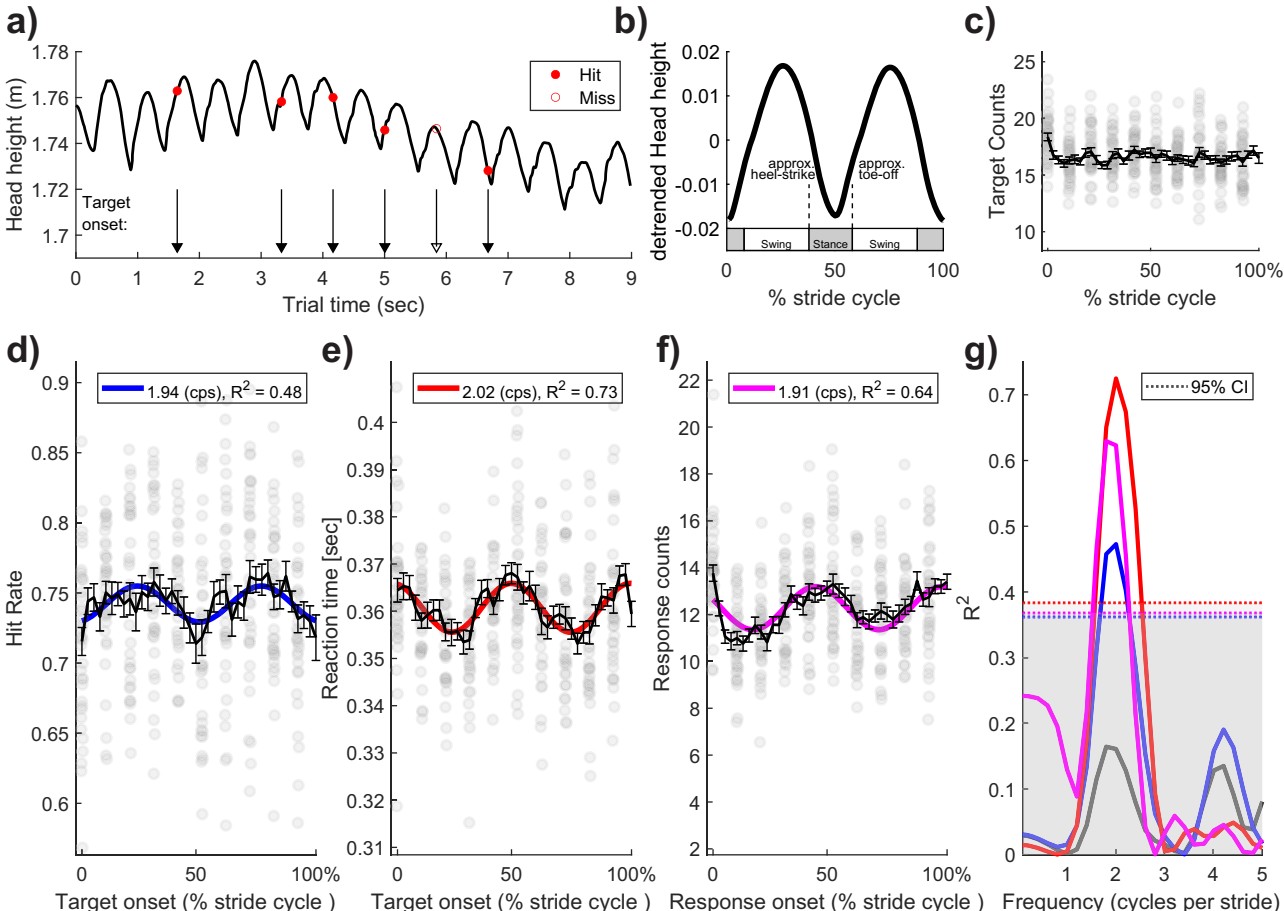

**Fig. 3 | Performance oscillates within the stride cycle. a** An example trial from one participant displaying the position-tracked vertical head height over time whose peaks and troughs are used to define the steps of a stride cycle. Target onsets are displayed with black arrows, and red markers indicate the behavioural outcome of the task (Hit, Miss). **b** To allow pooling of data over participants, each stride (two steps) was resampled to a normalised range of 1–100%. The approximate swing and stance phases of the stride cycle are displayed above the x-axis. **c** Target presentation density was approximately uniform over the stride cycle, validating the random-probing and stride epoching procedure. **d**–**f** Group level data (N = 36) show clear oscillations over the stride cycle for detection accuracy (hit rate), reaction time, and the likelihood of manual responses. The best-fitting first-order Fourier model is shown in each figure and approximated 2 cycles per stride

(i.e. the step rate). Individual data points (grey) are shown for 10% increments over the stride cycle. **g** Permutation testing of the best-fitting Fourier models. The observed data in **c**–**f** were fitted with a single-component Fourier model at all frequencies between 0.2 and 10 cycles per stride (in steps of 0.2 cps), and goodness-of-fit ($R^2$) was calculated. The prominent peaks at approximately 2 cps show that the best fits, whether for accuracy (blue), reaction times (red), or responses (magenta), occur at approximately 2 cycles per stride and far exceed the 95% permutation confidence interval calculated using the maximum test statistic across all frequencies per permutation (dotted lines and grey shading). These peaks did not co-occur with the above-chance fluctuations in the likelihood of target presentation, which is shown by the grey line.

the prevalence of significant oscillations in the 0.2–10 cps range per individual participant.

Consistent with the group-level results, we found oscillations at approximately 2 cps were the most prevalent in the sample. For example, for oscillations in accuracy, 12 of 36 participants had strong fits at approximately 2 cps that exceeded the 95% confidence interval of their shuffled data, rejecting the null at the participant level null-hypothesis significance test (NHST). Similar results were found for reaction time (n = 13/36 NHST at 1.5–2.5 cps) and response likelihood (n = 17/36 at 1.5–2.5 cps). Notably, a large proportion of our sample exhibited significant oscillations at higher frequencies, particularly at approximately 4 cps (accuracy n = 13/36; reaction times n = 14/36; responses n = 9/36 at 3.5–4.5 cps). Subsequent analysis indicated these individual-level oscillations, though idiosyncratic, were highly reliable. Few participants displayed significant oscillations in multiple frequency bands on any one measure (accuracy n = 4; reaction times n = 2, responses n = 3). Most displayed the same individual oscillation across all three measures, with few having different oscillations across performance measures (n = 3/36). Figure 4a–f displays a summary of

this data, showing the strength of individual participant Fourier fits at each frequency in cycles per stride. A small subset of participants demonstrated no significant oscillation on each measure (accuracy n = 6, reaction times, n = 3; responses n = 3 see Fig. 4g–i).

We proceeded by formally quantifying the population prevalence of these oscillations using Bayesian inference. Bayesian inference of population prevalence provides an estimate of an effect's within-participant replication probability and is particularly useful when examining effects which may be heterogeneous[27,28]. This method returns the maximum a posteriori (MAP) estimate of the population prevalence and a 95% confidence interval within which the MAP resides. For example, the prevalence estimate for oscillations in accuracy at 2 cps is 0.32 (MAP), with a 95% highest posterior density interval (HPDI) between [0.18, 0.49], indicating a 95% probability that the population prevalence of oscillations in accuracy at 2 cps exceeds 0.18. Similar results were found for reaction time (2 cps, MAP = 0.30, [0.16, 0.47]) and response likelihood (2 cps, MAP = 0.41, [0.26, 0.58]). Given the idiosyncratic yet reliable presence of oscillations, the Bayesian estimate for any oscillations in each of our measures over the

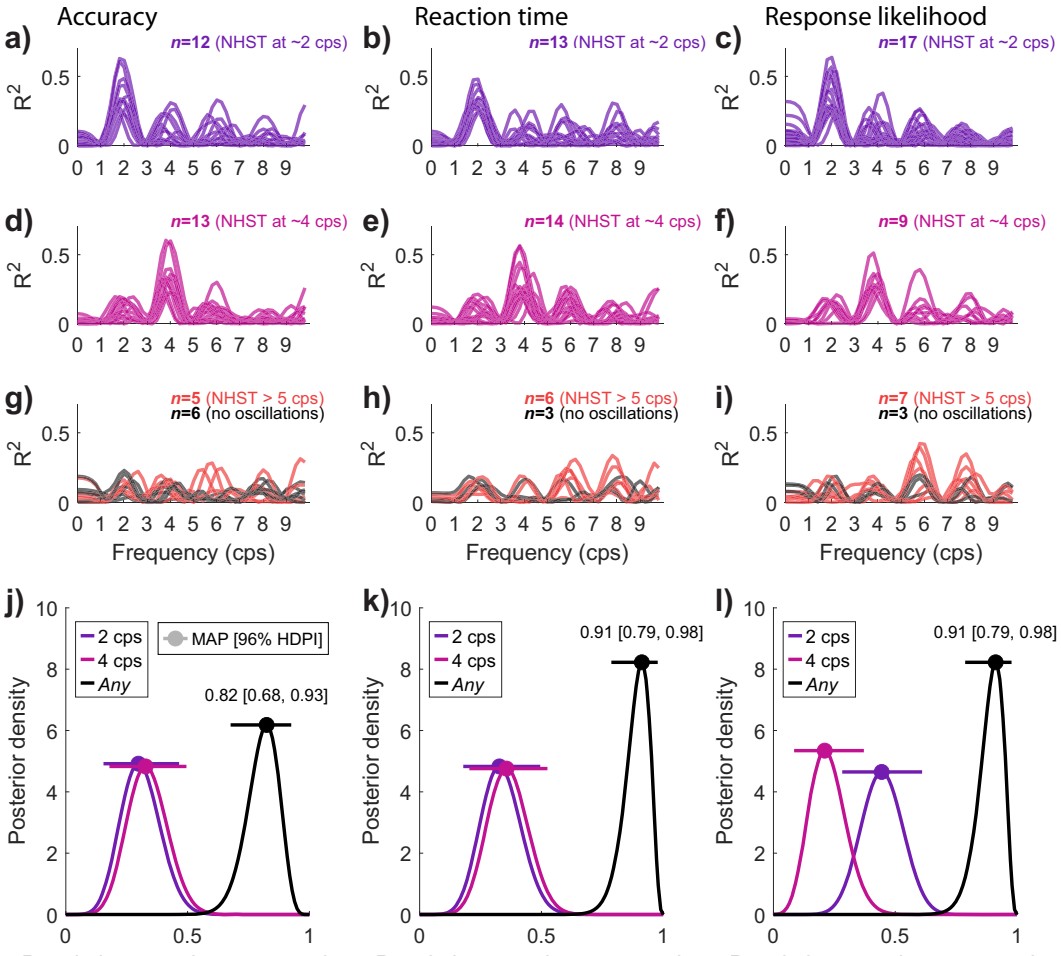

**Fig. 4 | Population prevalence of performance oscillations over the stride cycle.** **a** Subset of participants displaying significant oscillations in accuracy performance between 1.5 and 2.5 cycles per stride (cps). Each line is the Fourier fit strength to participant-level data between 0.2 and 10 cps (in steps of 0.2). The strength of the participant-level Fourier fit was compared to the 95% Confidence Interval of permuted data per participant (see Methods). For accuracy, 12/36 participants demonstrated significant oscillations at approximately 2 cps. **b**, **c** Display the same result for reaction time and response likelihood oscillations at 2 cps, respectively. **d–f** Independent subsets of participants with significant oscillations between 3.5 and 4.5 cps. **g–i** Remaining participants with significant oscillations either above 5 cps (n = 5, 6 and 7 out of 36 for accuracy, reaction time and responses, respectively), shown in red. Overlaid in dark grey are the fits for participants with no significant oscillation at any frequency (n = 6, 3, 3 for accuracy, reaction time and responses, respectively). **j–l** Bayesian estimates of population prevalence for oscillations at 2 cps (purple), 4 cps (pink), or any frequency in the range of 0.2–10 cps (black). Circular markers display the maximum a posteriori estimate of prevalence (MAP), and error bars display the 95% highest posterior density interval (HPDI) (see Methods).

range of 0.2–10 cps was high. For accuracy, 29/36 participants demonstrated oscillations in the 0.2–10 cps range (MAP = 0.82, [0.68, 0.93]), 33/36 participants displayed oscillations in reaction times (MAP = 0.91 [0.79, 0.98]), and 32/36 for oscillations in response likelihood (MAP = 0.88, [0.75, 0.96]). Figure 4j–l displays the prevalence estimates for oscillations at 2, 4, and any cycles per stride for each of our dependent measures.

We performed additional exploratory analyses to investigate whether these idiosyncratic frequencies correlated with raw stride cycle duration. One possibility is that these oscillations are not determined by stride cycle but a more general oscillator such as clock time, which, when resampled over different stride-lengths could result in the variability in peak cps we observe. As a toy example, a single oscillator at 4 Hz could be captured as a 2 cps or 4 cps oscillation within individuals with a 500 ms vs 1000 ms stride length, respectively. In our data, we observed no statistically significant correlation between the across-participant stride length and the strength of 2 cps fits, the strength of 4 cps fits, nor the frequency of maximum fit strength for any of our dependent variables (Supplementary Fig. 7). Thus, it is the phase of the stride cycle that matters, not the duration of each step. Consistent with our hypotheses, we found significant

negative correlations between the frequency of perceptual oscillations (in Hz) and stride cycle duration. Supplementary Fig. 8 demonstrates this covariation and that participants with shorter stride cycle durations had faster perceptual oscillations. As a result, when aligned by phase, clear group effects emerge. Together, these results indicate that significant oscillations in visual detection task performance occur in over 80% of our sample, depending on the phase of an individual's stride cycle, and with stable idiosyncratic frequencies in performance per participant.

## Oscillations in performance are phase-aligned across participants
Our previous analyses demonstrated that oscillations in visual detection performance are highly prevalent within our sample and occur at stable idiosyncratic frequencies per participant. We next inspected whether the phase of these oscillations would also be consistent across participants. For this analysis, we included only the subset of participants that displayed significant oscillations when compared to their shuffled likelihood, and below focus on those participants with oscillations in the 1.5–2.5 cps range (accuracy data n = 12, reaction times n = 13, response likelihood n = 17). The best-fitting Fourier model was

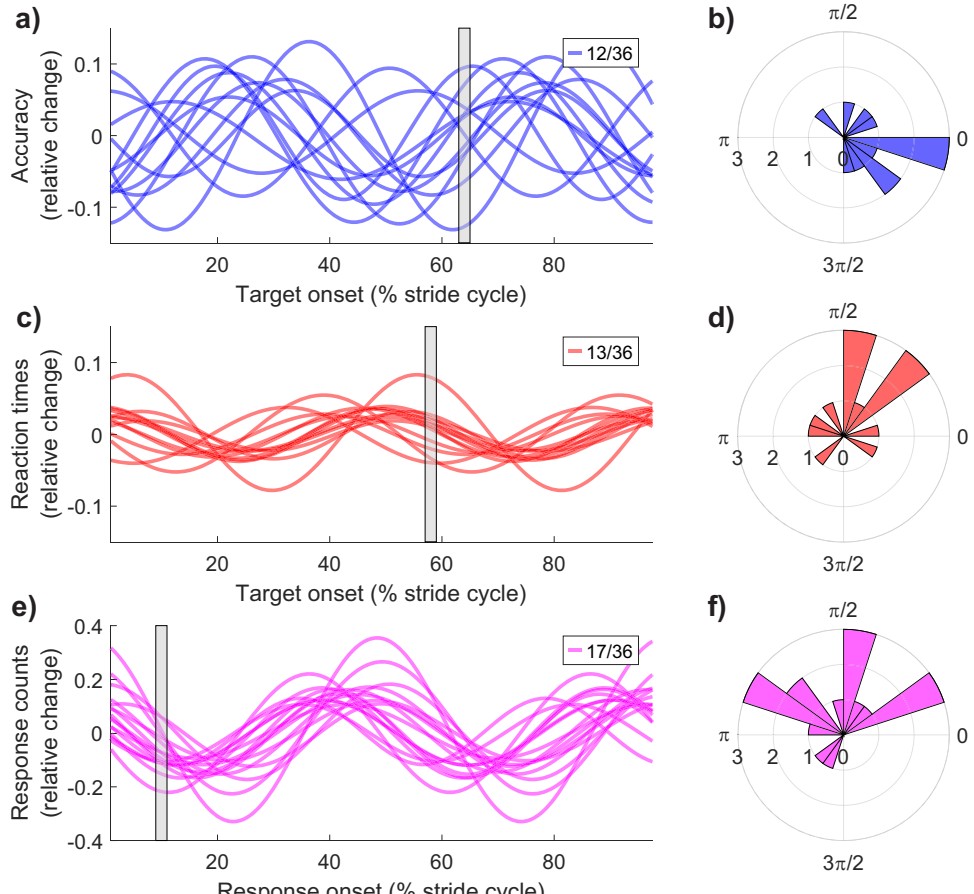

**Fig. 5 | Participant-level oscillations in performance are clustered in phase.**
**a** For participants with significant oscillations in accuracy at approximately 2 cycles per stride ($n = 12$), the Fourier series model at 2 cycles per stride is displayed. To ease the interpretation of differences in phase, participant-level data is expressed in relative change from their mean ((X-Mean)/Mean). The vertical grey shaded region notes the time at which the phase distribution in (**b**) corresponds to, and **b** displays the polar plot of phase across participants. **c**, **d** Reaction time data ($n = 13$).
**e**, **f** Response onset likelihood ($n = 17$).

calculated per participant, and the phase was retained for subsequent tests of non-uniformity. Figure 5 displays the phase results and shows the individual data are tightly clustered in the phase. Optimal accuracy (Fig. 5a) and reaction time (**C**) occur in the approximate swing phase of the stride cycle, while response likelihood (**E**) entrains to the time of heel strike. Rayleigh's test confirmed that each of these phase distributions were non-uniform (accuracy; $Z = 3.62$, $p = 0.023$, $k = 0.45$; reaction time; $Z = 3.59$, $p = 0.024$, $k = 0.47$; response $Z = 4.73$, $p = .007$, $k = 0.47$).

## Discussion

We investigated whether the natural phases of human locomotion alter performance on a simultaneous visual detection task. Participants walked along a straight path in a wireless position-tracked virtual reality environment and responded as quickly as possible to brief visual targets presented at random intervals as they walked. Offline analysis of head height data was used to define the start and end of each stride cycle and to determine the timing of the target relative to stride onset. Pooling over many trials and dividing the data into fine time bins, we confirmed that targets occurred uniformly across the stride cycle. However, analysing accuracy, reaction time and manual response likelihood revealed clear oscillations in all three performance measures that were systematically linked to the phase of locomotion. Further analyses of individual participants confirmed that these group-level performance oscillations were present in a large proportion of our sample, predominantly at either 2 or 4 cycles per stride (cps). While the frequency of 2 or 4 cps was idiosyncratic, there was a strong

consistency within participants so that the same frequency was seen for a given participant across the three performance measures.

Together, these results reveal systematic modulations of visual performance within the stride cycle. This work also validates our approach as a viable method that will enable further research into stride cycle modulations of perception and performance in vision and other sensory modalities.

What might underlie these oscillations in performance during locomotion? It is important to note that oscillations are a ubiquitous feature in neuroscience, in both the brain and behaviour[29]. Prior work has established that performance on a visual detection task waxes and wanes over time (see[30,31] for reviews). These behavioural oscillations are in the theta or low alpha range (4–10 Hz), and have been variously linked to the reallocation of attention[32–34] to perceptual sampling[35], or to decisional processes[36]. Often in these studies of behavioural oscillations, a salient transient stimulus is used to 'reset' the phase of ongoing oscillations[37,38] so that they can be probed from a known and repeatable temporal reference point. For example, this approach has been used to reveal oscillations in perceptual sensitivity in both vision and audition[36,39], and counterphase oscillations that have been linked to the reallocation of attention[29,40,41].

Behavioural oscillations can also be revealed by using an action at the start of the presentation period to reset the phase of neural oscillations. Actions as simple as reaching or making a button press are sufficient to provide a time-zero point from which underlying oscillations can be probed over time[42–45]. Our work, though, has extended this line of research to identify oscillations in performance that are

synchronised to the rhythmic action of locomotion. On average, our participant group exhibited best-fitting Fourier frequencies which spanned the range of 1.93–2.02 cycles per stride (Fig. 3d–f). This raises the possibility that the act of taking a step resets the phase of neural oscillations and creates a repeatable reference point that causes an underlying oscillation to be revealed at approximately the step rate. It remains to be seen whether the oscillations we have described co-occur with slower endogenous brain rhythms, as has been the case for previous work investigating the fast-scale oscillations at higher rates in performance linked to attention and perceptual sampling[46]. Similarly, reaching behaviour has been shown to reset the phase of endogenous oscillations[43,47,48] and the ballistic movements generating locomotion may reset or entrain neural oscillations relevant to behavioural performance[49].

One interpretation of the oscillations observed in perception and attention is that they arise naturally as a kind of reverberation frequency. On this view the period length reflects the time taken for signals to feed forward to higher-level areas and then feed back to sensory levels. A similar pattern could arise in locomotion at a lower frequency governed by the step rate. Locomotion obviously involves feedforward motor signals, but the control of locomotion is not possible without neural and mechanical feedback[50,51]. There are also numerous findings in rodents showing that locomotion elevates sensory cortical response gain generally and visual cortical activity specifically[13,15–17,52–54]. It is not known yet whether these condition-average elevations interact with the phases of the stride cycle, but this established link between locomotion and visual cortical activity raises a testable and more specific proposition that the stride cycle may modulate activity in visual cortex and thus produce oscillations in visual performance entrained to each step.

Also relevant to the present work are recent studies investigating brain–body coupling in the context of perception and motor commands. The rhythmic influences of the cardio-respiratory system, for example, are increasingly recognised to contribute to variability in perceptual task performance[55–64]. Our findings extend this line of work by showing how steady-state locomotion also rhythmically alters sensory performance and that these effects are highly prevalent and idiosyncratic. Notably, heart-rate[65–67] and respiration[68–70] also show coupling to walking behaviour–leaving open the possibility that the oscillations we describe may also be synchronised with the cardio-respiratory or other bodily systems[63]. One intriguing possibility is that the variations in idiosyncratic peak frequency we report are not coupled to stride cycle duration (cf. Supplementary Fig. 4) but cardio-respiratory exertion. Future studies can individually calibrate walking speeds, as well as experimentally decouple the cardio-respiratory cycle from the stride cycle (such as while carrying a backpack or during inclined walking), to investigate this potential confound.

Similarly, recent work has identified that the timing of blinks and saccades also entrain to the rhythm of footfall[23], and that faster perceptual fluctuations may be time-locked to saccade offset[71]. In our data, the occurrence of saccades and blinks was relatively sparse owing to the smooth-pursuit nature of our task, yet future work could quantify the relative contribution of saccades to these oscillations using tasks that evoke large eye movements–for example, by presenting targets that vary unpredictably in peripheral eccentricity to the left or right of fixation.

The relative timing of the modulations in visual task performance we report is also noteworthy for other reasons. Previous work has indicated how the subjectively smooth nature of walking is nevertheless supported by phasic periods of enhanced sensory demand. In particular, after toe-off in the early phase of the step cycle and prior to heel-strike, the predictability of head movements is at their lowest[72], leading to an increase in the weighting of vestibular signals to maintain balance and posture[73,74]. Indeed, it is well established that the evolution of locomotion required the widespread coordination of a host of bodily systems[75,76]. In animal models, a key result from this research is that although walking may appear smooth and continuous, there are phasic periods within each step cycle that ballistically determine an ongoing step trajectory (reviewed in[75,76]).

In summary, while an increasing body of work has compared stationary conditions to light exercise or continuous walking[7–11,77], we have demonstrated that reliable measurements can be made within the stride cycle. In the current paper, this has revealed clear changes in visual performance linked to the phase of the stride cycle. These findings open many research possibilities concerning, for example, how and where attention is allocated over the stride cycle, whether visual modulations occur uniformly over the visual field, and whether performance on auditory or tactile tasks will also modulate with locomotion.

## Methods

### Participants

This research complies with all relevant ethical regulations and was approved by the University of Sydney Human Research Ethics Committee (HREC 2021/048). We recruited 45 healthy volunteers via convenience sampling, 7 of whom were excluded for incomplete data collection that either resulted from wireless signal drop-out or hardware malfunction. One more was excluded on the basis of eye-movement data (detailed below), and another for an error in eye-movement calibration. The remaining 36 volunteers included 22 females (Mean age = 19.6, SD = 2.6) with normal or corrected to normal vision. No statistical method was used to predetermine the sample size. The experiments were not randomised as all participants completed the same design without unique group assignments, and as such, the experiments were not blinded to allocation during experiments and outcome assessment. Sex and/or gender were determined based on self-report and were not considered in the experimental design, which focused on within-participant modulations over the stride cycle. All participants were recruited from the University of Sydney undergraduate psychology cohort, provided informed consent prior to participation, and received either course credit or 20 AUD per hour of their time.

### Apparatus and virtual environment

The virtual environment was built in Unity (version 2020.3.14f1) incorporating the SteamVR Plugin (ver 2.7.3; SDK 1.14.15), on a DELL XPS 8950, with a 12th Gen Intel Core i7-12700K 3.60 GHz processor, running Microsoft Windows 11. The virtual environment consisted of an open simulated woodland, sparsely populated with trees, as we have used previously[26]. Participants walked along a simulated 9.5 m track wearing the HTC Vive Pro Eye with an integrated head-mounted display (HMD) and a wireless adapter kit (130 gram weight) and carried two wireless hand-held controllers. The HMD houses two $1440 \times 1600$-pixel (3.5″ diagonal) AMOLED screens with a 110° field of view refreshed at 90 Hz. The HMD also integrates Tobii eye-tracking technology, from which we recorded gaze origin and gaze direction information using the native SRanipal SDK (v1.1.0.1). During the experiment, the three-dimensional coordinates of the HMD, gaze-origin, gaze-direction and hand-held controllers were sampled at 90 Hz resolution, using five HTC Base Stations (v 2.0). Participant responses were collected from the wireless controllers and for self-pacing at the onset of each trial using a trigger button beneath the right index finger.

At the start of each trial, the starting position was indicated by a large red "X" on the ground. The red X positioned participants in line with a walking guide, which was a small cube (0.1 m on a side) at approximately waist height. The cube had a directional arrow on the superior face displaying the required direction of motion on walking trials or a stop signal on stationary trials. Positioned above the walking guide on each trial was a rectangular screen (35 cm width, 20 cm height). The average Euclidean distance between our

participants' HMD and the virtual screen was 52 cm, resulting in an average screen size of 37.2° × 21.8° of visual angle The vertical centre of the screen was calibrated to 80% of the height of each participant's standing HMD location. This height calibration standardised the approximate viewing angle to the screen across participants. In the centre of the screen was a circular annulus (11 cm diameter, 12.1°) which indicated the boundary of the region to monitor for potential target appearances. Within this circular region, targets were briefly flashed with a grayscale contrast that was titrated using an adaptive staircase to standardise performance across participants to approximately 75% (see "Staircase procedure" below). The screen colour was uniform grey (RGB 0.4, 0.4, 0.4), and the target shape was a small ellipse, 1.5 cm in length on its long axis (subtending 1.66°), tilted at ±45° from vertical.

## Procedure and task stimuli

Upon arrival, the participants read over the information sheet, became familiar with the testing area, and had the chance to ask questions before providing informed consent. They were given a brief overview of the wireless virtual reality equipment, hand controllers, and battery pack before beginning the first practice block which was always the stationary condition.

Each experiment contained 10 experimental blocks, with 20 trials per block. Each trial was 9 s in duration, spent either standing still or walking the 9.5 m distance at a steady speed, with the smooth duration and speed of travel set by the walking guide. On stationary trials, participants stood motionless for 9 s, while on walking trials, the walking guide progressed at a smooth speed of 1.1 m/s. The order of the first two blocks was pre-set (stationary, walking), after which the remaining 8 blocks were randomised (two stationary and six walking blocks). Through pilot testing, we observed that accuracy quickly settled to approximately 75% in both conditions due to our adaptive staircase procedure, and consequently increased the trial count for the walking condition to enable detailed stride cycle-based analyses, which was our main focus of interest (see "Performance relative to stride cycle" below).

Within each 9 s trial, we carefully spaced target presentation times to enable sequential responses per trial. A maximum of eight targets were presented within each trial, with a minimum inter-trial interval (ITI) of 0.8 s. Each target was 20 ms in duration with a titrated contrast determined by adaptive staircase (see "Staircase procedure" below). The spacing of targets was relative to predefined anchor points (e.g. target 1 at 1.1 s, target 2 at 2.3 s …), with an additional jitter (±0.5 s) to decrease the predictability of target onset within each trial (while ensuring the minimum ITI was satisfied). To further decrease the predictability of target onset times, each target had a 10% chance of being withheld from presentation, resulting in a range of actual target presentation counts per trial per participant (Min = 0, Max = 8, M = 6.99, SD = 0.07), as well as increased range of inter-trial intervals overall (M = 0.92 s, SD = 0.10 s). This procedure resulted in an average of 507.66 stationary targets per participant (SD = 68.87) and 884.13 walking targets per participant (SD = 60.42). Importantly, this target presentation methodology resulted in an approximately uniform distribution of target onsets over the stride cycle (Fig. 3).

During each trial, the target location was also manipulated to increase engagement and attention to task. The circular annulus defining the boundary for target appearances slowly drifted within the confines of the background screen (speed range of 0.1–0.6 m/s). Targets could appear anywhere within this circular boundary, excluding regions that overlapped with the circular annulus. Supplementary Fig. 1 displays a summary of all target positions throughout the experiment.

## Staircase procedure

Our adaptive staircase was an implementation of the QUEST procedure[78], programmed in C# for the Unity virtual environment.

Target contrast was adjusted on a single-trial basis separately for stationary and walking blocks. For both staircases, we set the initial target contrast to 0.45 (the background screen was 0.4). We set the initial slope ($\beta$) parameter for QUEST at 3.5, standard deviation to 2, with a lapse rate of 0.01 and floor at 0.5. Both staircases had a resolution of 0.004, ranging in contrast from 0.4 to 1.

The staircase procedure began after the first three trials of the first practice block, which were omitted from further analysis. During these first three trials, target contrast was set at the suprathreshold starting parameter of 0.45 to familiarise participants with an easier version of the task. For the remainder of the experiment, we additionally manipulated target contrast by perturbing the QUEST staircase on a target-by-target basis. As the QUEST procedure quickly converged to a contrast value that approximated 75% accuracy, we selected each target contrast from one of seven positions within the 25th to 75th percentile of the prior probability density function. These positions were selected to enable psychometric fits to participant data and spaced at percentiles [25, 32.25, 37.5, 50, 62.5, 68.75, 75]. For simplicity, we refer to these locations as [Q-3, Q-2, Q-1, Quest mean, Q + 1, Q + 2, Q + 3] henceforth.

## Psychometric fits

Using the dispersion of contrast intervals described above, we modelled psychometric fits to participant data using a cumulative normal distribution, implemented in the psignifit toolbox[79]. From each function, we retained the 50% threshold, fit width (difference between the 5th and 95th percentile of the data), and slope parameter. Figure 2 displays the average of participant fits during stationary and walking conditions.

## Eye movement data

We collected eye movements using the SRanpal SDK and integrated Tobii eye-tracking technology. Eye movements were calibrated at the start of each experiment using a 4-point calibration procedure following the manufacturer's settings. Additional calibration was performed after any change to the headset (such as after breaks or between blocks). We recorded 3D coordinates of eye position and gaze direction at each frame of all trials at 90 Hz resolution and performed the following preprocessing steps to identify and omit blinks from our analysis.

First, whole-trial time-series of the eye position and gaze direction data were tested for outliers. Blinks were identified using gaze direction data on the basis of data discontinuities that exceeded ±0.8 m. Raw time-series data was then interpolated from −200 ms before to +200 ms after each blink, using a modified Akima piecewise cubic Hermite interpolation. Identical windows were interpolated for the gaze origin and gaze direction time series on both axes. Overall, a low number of blinks were detected, with a low participant average per 9 s trial (M = 1.12, SD = 1.08, range 0.035–4.05). Consequently, there was a low incidence of blinks overlapping with target onset (within ±200 ms; M = 0.07, SD = 0.13, range 0–0.63). After interpolation, the vertical eye position data was linearly detrended to enable averaging across participants of different heights. Next, the cartesian coordinates of gaze direction at target onset were converted to polar coordinates, centred on the target's location. All target events that appeared when the gaze location exceeded 12.1° of visual angle from the target's position were excluded from further analysis. This procedure was in place to ensure that participants were gazing within the approximate region of the circular annulus when targets were presented. This procedure identified one participant for exclusion based on a large number of missed targets but otherwise demonstrated that participants faithfully tracked the circular annulus with their gaze, as a low number of targets were rejected on average per participant (M = 57.47, SE = 30.2, approximately 4% of targets).

## Gait extraction from head-position data

We estimated the phases of the stride cycle by applying a peak-detection algorithm to the time series of head position data. As walking results in near sinusoidal changes in the vertical centre of mass over time, troughs on the vertical axis of head position correspond to when both feet are placed on the ground during the double support stance phase[80–83]. We epoched all individual steps based on these troughs and normalised step lengths for stride cycle analysis by resampling the time-series data to 200 points (1–100% cycle completion, in steps of 0.5%). We also visually inspected individual head-position time series during pre-processing to identify trials for exclusion, based on wireless signal drop-out or poor performance of the peak-detection algorithm. Over all participants, an average of 1.1 trials (SD = 1.3) were rejected in this manner (range 0–5).

## Performance relative to the stride cycle

Our main analysis compared performance on the visual detection task relative to position in the simultaneously occurring stride cycle. To accomplish this, all target onsets were allocated to the percentile (from 1-100 %) at which the target occurred in the simultaneous step (Supplementary Fig. 2) or stride (Fig. 3). For single-step analyses, we averaged performance within 20 linearly spaced bins (1–5%, 6–10% etc., with zero overlap). For the stride cycle, the same procedure was performed over 40 linearly spaced bins.

We analysed detection accuracy and reaction time relative to the point of target onset within each stride cycle, as well as response likelihood (the frequency of self-initiated responses to targets). For each dependent variable, we tested for a significant oscillation via a two-step procedure. At the group level, we fit a sequence of Fourier series models within a forced frequency range, stepping from 0.2 to 10 cycles per stride in 0.2 increments, using MATLAB's curve fitting toolbox and the equation:

$$f(t) = a0 + a1 \times \cos(wt) + b1 \times \sin(wt) = A\cos(wt + \phi) \quad (1)$$

Where $w$ is the periodicity (cycles per stride) $t$ is time, $a1$ and $b1$ are cosine and sine coefficients, and $a0$ is a constant. The resulting sinusoidal fit has amplitude $A$ and phase $\phi$. The routine implements a non-linear least squares method that minimises the summed squares of the residuals over 400 iterations. For each forced fit at each frequency, we retained the goodness of fit ($R^2$) as our critical value. Figure 3 visualises the goodness of fit for each Fourier series model on the observed group-level data, with clear peaks at approximately 2 and 4 cycles per stride. Next, we performed a non-parametric shuffling procedure to assess the likelihood of these fits occurring by chance. Specifically, for each participant, we shuffled the data allocated to each percentile bin at random (without replacement), before repeating the group-level analysis and fitting procedure described above. This shuffling procedure was repeated 1000 times, and the distribution of $R^2$ values at each frequency from shuffled data served as the null distribution for the strength of sinusoidal oscillations at chance—when the temporal order of the stride cycle had been destroyed via shuffling. We compared the observed $R^2$ value to a null distribution created by retaining the maximum $R^2$ value across all frequencies per permutation and interpreted the presence of a significant oscillation when the observed $R^2$ value exceeded this critical value (effectively controlling for multiple comparisons).

We performed a series of additional analyses to further explore these oscillations in performance. We note that our main results of oscillations in performance also occur at the minimal width of 100 bins per step cycle. Supplementary Fig. 2 also displays that the result is present in single-step cycle analyses and that clear oscillations are present without any averaging (i.e. individual bins for targets at 0.5% increments of the stride cycle). We next examined whether mechanical artefacts may be contributing to these effects.

Given the decrease in detection accuracy around the time of footfall (during the loading phase of the stride cycle), one specific possibility is that the impact of footfall drives a mechanical distortion through the HMD. We hypothesised that if footfall was introducing a mechanical perturbation to the HMD and subsequent visual display, then the time series of pupil origin should show an increase in variability at this same time. However, no statistically significant increase in the variability of pupil origin was observed, indicating minimal displacement of the HMD during this phase of the stride cycle. Instead, we see clear evidence of an active compensation of gaze direction to account for changes in head position, as has been reported in other studies[24,25]. Supplementary Fig. 3 displays this result. We also note that transient mechanical distortions cannot account for the higher frequency oscillations in a large portion of our sample (see Fig. 4) or the existence of counterphase modulations in performance in a subset of participants (i.e. separated from the time of heel-strike (see Fig. 5).

We also repeated our main analysis to test for oscillations in performance when aligning to trial-onset, using clock-time as the basis of our Fourier fits. This allowed us to test for oscillations in standing data, as well as to compare whether the goodness-of-fits obtained when testing cycles per stride exceeded those we could measure when testing cycles per Hz. The results of this analysis are displayed in Supplementary Figs. 5 and 6.

## Participant-level analysis and Bayesian population prevalence

We additionally performed the same fitting procedure on individual participant data to test for the presence of a significant oscillation between 0.2 and 10 cycles per stride per participant. Performing participant-level null-hypothesis significance testing (NHST) also enabled a test of Bayesian inference of population prevalence—a recently proposed alternative to the population average effect size[27]. Bayesian inference of the population prevalence has several advantages compared to the group-level binary result of a NHST[27,28] and is particularly powerful when investigating effects that may be heterogeneous within a particular sample. After quantifying the proportion of participants that have a significant effect, this method estimates the population prevalence (within-participant replication probability) and quantifies the maximum a posteriori (MAP) estimate from this posterior distribution. Importantly, Bayesian prevalence provides both the most likely value of the population parameter, but also an explicit estimate of uncertainty within which the population prevalence resides. Here, following the recommendations of Ince et al.[28], we report the MAP and the 95% highest posterior density intervals (HPDI). These intervals provide the range within which the specified population parameter resides with 95% probability.

From the individual-level fits, we also compared the phase distributions of oscillations across participants and tested for phase clustering using Rayleigh's test of non-uniformity[84]. From these tests, we report the test statistic ($Z$), $p$-value and circular variance ($k$). Figure 5 displays the individual level fits (when significant) and phase distributions for performance oscillations at 2 cycles per stride.

## Data analysis

All analyses were performed in MATLAB (ver 2022a) or JASP (0.16.3.0). For each $t$-test and ANOVA reported, the data met the assumptions of normality and assumptions of equal variances unless otherwise stated. All statistical tests used were two-tailed unless otherwise stated. Psychometric fits were performed using the Psignifit toolbox[79]. We have implemented ColorBrewer[85] and Perceptually Uniform Colormaps (https://bids.github.io/colormap/) to aid in data visualisation. The 3D avatar was placed in the Unity environment for illustration purposes only and is available from www.passervr.com.

## Reporting summary

Further information on research design is available in the Nature Portfolio Reporting Summary linked to this article.

## Data availability

The raw and processed behavioural data generated in this study have been deposited in a public database (https://doi.org/10.17605/OSF.IO/8DJTQ) on the Open Science Framework and can be accessed using the link https://osf.io/8djtq/.

## Code availability

Analysis codes to reproduce the analyses and main figures are available in the public repository (https://doi.org/10.17605/OSF.IO/8DJTQ) and can be accessed using the link (https://osf.io/8djtq/).

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

## Acknowledgements

M.J.D., F.V. and D.A. were supported by the Australian Research Council (DP210101691).

## Author contributions

M.J.D. and D.A. designed the research. M.J.D. completed the research, performed all analyses and wrote the paper. M.J.D., D.A. and F.A.J.V. revised the final paper.

## Competing interests

The authors declare no competing interests.
