## [Peer Review File · Nature Communications]

Walking modulates visual detection performance according to stride-cycle phaseREVIEWER COMMENTS

Reviewer #1 (Remarks to the Author):

The manuscript "Walking entrains unique oscillations in performance on a visual detection task" by Matthew J. Davidson, Frans Verstraten, and David Alais describes a fluctuation in performance (RT, hit rate, response likelihood) that is locked to the step cycle in various but stable frequencies.

I must say, I have seldomly reviewed such a concisely written paper and, albeit I cannot fully judge the Bayesian approach, methods and analysis are very convincing. The insight, that our perception is not a solitary process but embedded in our ongoing natural movement is a very compelling concept and it is great to see such strong evidence as presented here.

My questions and suggestions are detailed below.

My main point is that the possible connection between percept and eye movements is not considered. If the step cycle is linked to eye movement behavior (i.e. modulation of blink and saccade rate by the step cycle, Cao et al, 2020) and eye movements are linked to perceptual cycles, as shown by Hogendoorn (2016; *J Cognitive Neurosci* 28:1625-1635), the perpetual fluctuation within a step cycle could be based on the timing of blinks and saccades within the cycle. Unfortunately, your eye-tracking signal sampled at 90 Hz can likely not detect smaller saccades. An additional EOG measurement could have allowed the relevant analysis. Nevertheless, as the main conclusion is that perception fluctuates with respect to the step cycle during walking, I think it is ok to only later pin down if this is achieved via an adaptation of eye movements or other, more direct processes. However, I think it is very important to discuss this possible connection.

You describe your blink detection as (ln 471 ff) : "...whole-trial time-series of the eye-position and eye-origin data were tested for outliers. Blinks were identified on the basis of outliers in the first-derivative of these time series that exceeded ± 0.4 m." I take it that you detected blinks on the gaze data. How can you make sure you did not also exclude larger saccades? Does the system not provide pupil data? How many target events were during a blink detection period?

You write (ln 476 ff): "the cartesian coordinates of gaze direction at target onset were converted to polar coordinates, centred on the target's location." Can you specify at what time period the gaze needed to be within 2 degrees of the target location? The target was unpredictable in time and space (where in the circle it was presented), so it seems surprising that subjects are able at all to focus their gaze on the target at target appearance. Please clarify. I understand that you wanted to show that the participants are gazing at the circle in which the target could disappear, but why at the target? Additionally, you write that the diameter of the circular annulus in which the target could appear was 11cm but don't give an approximation of visual angle. But as you later state that the 1.5 cm target converts to 0.03 degree visual angle one would assume the annulus was 0.22 degree visual angle. Degree of target and annulus seem too small. Is the calculation correct?

The connection to internally fluctuating perceptual and attentional processes is very nice. You might also want to consider the work by Landau, A. N., & Fries, P. (2012). Attention samples stimuli rhythmically. *Current biology*, 22(11), 1000-1004.

Signed: Barbara Haendel

Reviewer #2 (Remarks to the Author):

This is a nice study investigating how walking entrains oscillations in performance on a visual detection task. In the study, it was tested whether behavioral performance in a near threshold detection task varies along the stride cycle. This is a novel and innovative idea that combines natural physiological

behaviour (here walking) with cognitive performance and have thus a change yielding new conceptual understanding of the links between mind and body.

In the experiments, participants were either standing still or walking with a constant speed, while performing a near threshold detection task. The characterization of the performance in the near threshold detection task was comprehensive. It was found that performance oscillates within the stride-cycle so that for accuracy, 14/36 participants demonstrated significant oscillations at approximately 2 cps or faster and that participant performance was clustered in phase. However, the weakness of the study was that walking was fixed to 1.1 m/s and thus stride cycles length was always constant. As there was no variability in the cycle length that could be utilized in the analysis, it is not possible to draw firm conclusions that stride cycle would be the basis oscillations in behavioral performance. Even more so, as the performance oscillations were not compared between walking and standing still.

Major

- The performance in near threshold detection task was evaluated as oscillations in the Accuracy and Reaction times. It was found that both individual level and group level data showed clear oscillations in the accuracy with the stride cycle. However, as the walking was fixed to 1.1 m/s, the cycle length of stride was fixed and there was no variance in the stride cycle length that could be utilized in the analysis. To obtain more conclusive evidence for that stride cycle drives perceptual oscillations, it should be shown that perceptual oscillations depend on the stride cycle length. If the participants would walk faster or slower, would also the performance begin to oscillate in different frequency? I think it is not possible to draw firm conclusions on that performance fluctuations are caused by stride cycle, if such data is not provided,

- The main manipulation in the experiment was that participant spent either standing still or walked fixed to 1.1 m/s. Yet, there are no utilization of this main manipulation in the analysis of perceptual oscillation, but all analysis are about the stride length influencing performance. As this was not explicitly stated anywhere (neither in figures, figure legends or main text), I might also be wrong. To establish that walking (not stride cycle, as it was not manipulated) is associated with perceptual oscillations, it should be compared against standing still which should so any perceptual oscillations if they are to arise from walking.

- The participants were to walk a 5 metre distance at a steady speed, with the smooth duration and speed of travel set by the walking guide. How was this controlled? Is it possible that participants did not obey this instruction and walked in a different speed? Data of the stride cycle durations at the group level should be presented if the participants were not physically bound to walk with a fixed speed e.g. with a treadmill. If there is variability in stride cycle duration, this could be utilized to test if the perceptual oscillations vary as a function stride cycle duration.

Minor

- The set-up is quite difficult to understand from the method section. A schematic illustration of the setup would be helpful to understand how participants were positioned and how they performed the experiment.

- Also, the description of the near threshold detection task was rather superficial and should be more detailed. What is the stimulus length, timing, color and luminance and what is the inter-trial interval?

- The fig 5 nicely illustrates that performance oscillations are clustered in phase. Could you better explain to what phase in stride cycle, these phase distributions corresponds to?

Reviewer #3 (Remarks to the Author):

This paper presents an exciting and to my knowledge novel result about how visual perception may be influenced by the gait cycle during natural walking. Using a VR system they are able to present a laboratory-style visual detection task during short segments of natural walking. Their results show that hit-rate, reaction time and response counts appear to follow a sinusoidal modulation over the stride cycle. To me this seems highly significant and would be of broad interest. I have some comments on the statistical methodology which I outline below.

Major comments

I have two concerns about the statistical analysis presented. The first is the lack of correction for multiple comparisons and the second is about the shuffling procedure used to generate samples from a null model.

Multiple Comparison Correction (over frequency models considered)

The key statistical analysis of the paper quantifies how well the pattern of behavioural effects can be fit by a Fourier oscillatory model of different frequencies. This leads to the problem of multiple comparisons – a number of statistical inferences are performed, one for each frequency considered. This inflates the false positive rate of the test. When performing a single test at $p < 0.05$ the false positive rate is controlled at 0.05. When performing 50 tests (as for the 50 frequency values considered here), the overall false positive rate is now $1 - \text{binocdf}(0, 50, 0.05) = 0.92$. This means, in an experiment fitting 50 frequency models, would expect at least one frequency to provide a significant fit in 92% of experiments carried out, even if the overall null was true at every frequency (i.e. there was no oscillatory effect at all). This applies both for inferences at the group level, but particularly for the within-participant analysis. In this case one might argue they are only testing 2 specific frequencies (2 and 4 cps), but these have obviously been chosen after observing the data, and the “any” effect inference is incorrect (because the false positive rate of any effect is not 0.05). A way to address this would be use the maximum statistics procedure with the shuffling approach they currently use (but see next point). For this, instead of taking the 95th percentile of the 1000 shuffled permutations *for each frequency separately*, they should take the maximum over frequencies, for each permutation, and then look at the 95th percentile of this maximum. This gives you a value of R^2 which you might expect to see in one of the 50 frequencies considered, under the null hypothesis, so this threshold controls the family-wise error rate of the cross-frequency test procedure to be 0.05 (which is required for prevalence estimation of the “any” effect, and protects against double dipping in terms of reporting prevalence of 2/4 cps).

Shuffling Procedure

Notwithstanding the multiple comparisons across frequencies as outlined above, to determine the significance of each specific frequency the authors shuffle the data across time to create a surrogate distribution. This removes both periodic and aperiodic structure as noted in this important and relevant paper: <https://www.nature.com/articles/s41562-022-01364-0>

The situation here is slightly different to the one described in that paper (which is usually about ongoing dense sampling, rather than studying entrainment to a separate fixed cycle, here stride). But I think the key point remains. It would be better to use a shuffling procedure that preserves the temporal autocorrelation of the data. One way to do this would be rather than shuffling trial events across the stride cycle, randomly shuffle the stride-cycle onsets across the experiment and repeat the analysis.

I also have a concern about the overall interpretation. While, notwithstanding the above points I am convinced that there is an oscillatory pattern in the visual behavioural variables considered, I miss any kind of control analysis to demonstrate that these are locked to the stride cycle specifically, rather than being a less-specific oscillatory phenomenon. For example, I would want to see some sort of control, perhaps looking at performance with respect to phase of an oscillation aligned to trial onset, would an oscillatory model fit there and would the effects be weaker than the stride alignment. Is

stride phase a better reference for behavioural modulation than say clock-phase locked to trial onset. I think related to this there could be more explicit discussion of potential confounds in the discussion. For example, the authors mention that respiration can be linked to the gait cycle, but they don't explicitly note how that could be a confound here. How could future work look to determine whether this visual perception is more strongly influenced by the respiratory cycle or the gait cycle (presumably those would have to be decoupled experimentally). Similarly, Supplementary Figure 3 shows smooth change in gaze origin, which follows an sinusoidal cycle over a single step. But this shows exactly the same patten as the visual behaviour, so is this not a confound for that?

Minor comments

Figure 3: why use a 0-100% scale rather than an explicit angular scale (to make clear the circular nature of the range), either radians or degrees.

L187: the 4cps is already visible in the group mean Fig 3d, so perhaps this could be commented on.

There doesn't seem to be much difference between the two strides of a cycle (I guess left and right), so perhaps they would have more statistical power (noting the extra controls needed above) looking at phase over a single leg movement, ignoring whether it is left or right. Difference between left and right stride could be tested explicitly. Single-step analysis is reference on line 503, but I don't see the results featured very prominently (not sure where these single-step analyses are featured).

In discussion, could maybe talk about more about neural oscillations. Could the heel strike or some other discrete part of the stide cycle be causing a phase-reset, would these fit into known behavioural oscillations.

REVIEWER COMMENTS

Reviewer #1 (Remarks to the Author):

The manuscript "Walking entrains unique oscillations in performance on a visual detection task" by Matthew J. Davidson, Frans Verstraten, and David Alais describes a fluctuation in performance (RT, hit rate, response likelihood) that is locked to the step cycle in various but stable frequencies.

I must say, I have seldomly reviewed such a concisely written paper and, albeit I cannot fully judge the Bayesian approach, methods and analysis are very convincing.

The insight, that our perception is not a solitary process but embedded in our ongoing natural movement is a **very compelling concept and it is great to see such strong evidence as presented here.**

- Thank you kindly for this glowing summary of our work.

My questions and suggestions are detailed below.

My main point is that the possible connection between percept and eye movements is not considered. If the step cycle is linked to eye movement behavior (i.e. modulation of blink and saccade rate by the step cycle, Cao et al, 2020) and eye movements are linked to perceptual cycles, as shown by Hogendoorn (2016; J Cognitive Neurosci 28:1625-1635), **the perpetual fluctuation within a step cycle could be based on the timing of blinks and saccades within the cycle.** Unfortunately, your eye-tracking signal sampled at 90 Hz can likely not detect smaller saccades. An additional EOG measurement could have allowed the relevant analysis. Nevertheless, **as the main conclusion is that perception fluctuates with respect to the step cycle during walking, I think it is ok to only later pin down if this is achieved via an adaptation of eye movements or other, more direct processes. However, I think it is very important to discuss this possible connection.**

- We agree that eye-movements could contribute to perceptual fluctuations during walking. We have quantified their contribution to this dataset and find they had minimal influence on the results.
- Specifically, our new analyses show that the number of saccades detected per 9 second trial was very low. Indeed, there was an average of 1.12 saccades per 9-s trial (SD= 1.08, range .03 – 4 per trial per participant). The reason for this low number is that subjects performed a smooth pursuit task as they walked (spatially tracking the circular place-holder within which the probe stimulus appeared). By smoothly pursuing the slowly moving circle, saccades were minimised.
- Similarly, the number of targets presented within the temporal range of blinks was also very low (± 200 ms, M per trial = .07, SD = .13, range 0 – .63). This low number is likely because participants were encouraged to withhold blinks until

the end of each 9 second trial, and during the trial were engaged in the smooth-pursuit task. We have included these summary statistics in our revised Methods:

Lines 546-549

Overall, a low number of blinks were detected overall, with a low participant average per 9 second trial ($M=1.12$, $SD= 1.08$, range .035 – 4.05). Consequently, there was a very low incidence of blinks overlapping with target onset (within ± 200 ms; $M = .07$, $SD=.13$, range 0–0.63).

- As our main contribution here is to demonstrate that perceptual fluctuations exist, and that they are time-locked to the stride-cycle, we have expanded on the possible role of eye-movements and directions for future research in our revised discussion:

Lines 391-397

Similarly, recent work has identified that the timing of blinks and saccades also entrains to the rhythm of footfall (Cao et al, 2020), and that faster perceptual fluctuations may be time-locked to saccade offset (Hogendoorn, 2016). In our data the occurrence of saccades and blinks were relatively sparse owing to the smooth-pursuit nature of our task, yet future work could quantify the relative contribution of saccades to these oscillations using tasks that evoke large eye movements – for example, by presenting targets that vary unpredictably in peripheral eccentricity to the left or right of fixation.

*You describe your blink detection as (ln 471 ff) : "...whole-trial time-series of the eye-position and eye-origin data were tested for outliers. **Blinks were identified on the basis of outliers in the first-derivative of these time series that exceeded ± 0.4 m.**" I take it that you detected blinks on the gaze data. How can you make sure you did not also exclude larger saccades? Does the system not provide pupil data? **How many target events were during a blink detection period?***

- Blinks were indeed detected on the basis of gaze data (pupil data was not recorded), and we acknowledge our previous criteria may have included large saccades.
- We have now re-performed our blink-detection analysis using more conservative criteria. As blinks are clearly identifiable in the gaze data as data discontinuities that exceed $\sim \pm 0.8$ m (example in *Review Figure 1* below), we now identify and interpolate all blinks which exceed this threshold. The data reported above regarding blink counts and target overlap is with this revised threshold (targets within ± 200 ms of a blink; $M = .07$, $SD=.13$, range 0–0.63).

Review figure 1: Example blink and saccade detection. (top panel) An example trial (participant 1, trial 61), showing the cleaned time-series of gaze-direction data (X coordinates in red, Y coordinates in blue). The blink identified is shaded in grey, with gaze-data prior to interpolation shown in black. This blink satisfied the criteria of a discontinuity exceeding ± 0.8 m. **(bottom panel)** Red and blue time-series plot the velocity of the gaze directions from the top panel. Automatic thresholds for saccade detection are shown with broken lines, and are calculated separately based on the SD of direction data on the vertical and horizontal axes. One saccade was identified that exceeded the threshold for Vertical velocity changes (shown with a red circle). Note we did not consider eye data in the first and last second of each trial when calculating thresholds or detecting saccades since these were outside of target presentation windows.

- We have now updated our methods with respect to blink detection:

Lines 541-546

“First, whole-trial time-series of the eye-position and gaze direction data were tested for outliers. Blinks were identified using gaze direction data on the basis of data discontinuities that exceeded ± 0.8 m. Raw time-series data was then interpolated from -200 ms before to +200 ms after each blink, using a modified Akima piecewise cubic Hermite interpolation. Identical windows were interpolated for the gaze origin and gaze direction time-series on both axes”

You write (ln 476 ff): “the cartesian coordinates of gaze direction at target onset were converted to polar coordinates, centred on the target’s location.” **Can you specify at what time period the gaze needed to be within 2 degrees of the target location?** The target was unpredictable in time and space (where in the circle it was presented), so it seems surprising that subjects are able at all to focus their gaze on the target at target appearance. **Please clarify. I understand that you wanted to show that the participants are gazing at the circle in which the target could disappear, but why at the target?**

- This and the next point are related - and we thank you for the opportunity to correct (and clarify) this criteria. First, we have now amended a previous error in our degrees of visual angle calculations.
- The targets were small (~1.7 dva), and could appear anywhere within the circular annulus (diameter ~12 dva).
- We wanted to demonstrate that participants were tracking the circular annulus, and indeed, they could not track the target location between onsets.
- After amending our dva error, we have re-analysed our dataset with a more liberal criterion. We now only remove targets from analysis if the difference between gaze direction and the target exceeds the diameter of the annulus (12 dva) at target onset.
- All our statistics and figures have been updated, and we note that our main results are robust to these changes.
- We acknowledge that some participants may still have been able to see the target in their periphery, but we prefer to remove this variability. Subsequent experiments (in preparation) are being performed to compare the effect of stride-cycle oscillations in the centre vs periphery with a balanced design.
- We have amended the corresponding section of our methods accordingly:

Lines 554-560

“Next, the cartesian coordinates of gaze direction at target onset were converted to polar coordinates, centred on the target’s location. All target events that appeared when gaze location exceeded 12.1 degrees of visual angle from the target’s position were excluded from further analysis. This procedure was in place to ensure that participants were gazing within the approximate region of the circular annulus when targets were presented. This procedure identified one participant for exclusion based on a large number of missed targets, but otherwise demonstrated that participants faithfully tracked the circular annulus with their gaze, as a low number of targets were rejected on average per participant overall ($M = 57.47$, $SE = 30.2$, approximately 4% of targets).

Additionally, you write that the diameter of the circular annulus in which the target could appear was 11 cm but don't give an approximation of visual angle. But as you later state that the 1.5 cm target converts to 0.03 degree visual angle one would assume the annulus was 0.22 degree visual angle. Degree of target and annulus seem too small. **Is the calculation correct?**

- This was incorrect, and the updated values based on the average viewing distance from the screen are target (1.66 dva), and annulus diameter (12.1 dva).
- These new values have been updated in our methods.

The connection to internally fluctuating perceptual and attentional processes is very nice. **You might also want to consider the work by Landau, A. N., & Fries, P. (2012). Attention samples stimuli rhythmically. *Current biology*, 22(11), 1000-1004.**

- We have added this citation to our discussion, and expanded the link to attentional sampling.

Lines X_X

“These behavioural oscillations are in the theta or low alpha range (4 –10 Hz), and have been variously linked to the reallocation of attention (VanRullen 2018; VanRullen et al. 2007; Landau and Fries 2012) to perceptual sampling (VanRullen, 2015), or to decisional processes (Zhang et al., 2019). Often in these studies of behavioural oscillations, a salient transient stimulus is used to ‘reset’ the phase of ongoing oscillations (Davidson et al., 2018; Romei et al., 2012) so that they can be probed from a known and repeatable temporal reference point. For example, this approach has been used to reveal oscillations in perceptual sensitivity in both vision and audition (Ho et al., 2017; Zhang et al., 2019), and counterphase oscillations in performance at separate locations that have been linked to the reallocation of attention (Fiebelkorn and Kastner 2019; Fiebelkorn et al. 2013; Landau et al. 2015).”

Signed: Barbara Haendel

Reviewer #2 (Remarks to the Author):

This is a nice study investigating how walking entrains oscillations in performance on a visual detection task. In the study, it was tested whether behavioral performance in a near threshold detection task varies along the stride cycle. **This is a novel and innovative idea** that combines natural physiological behaviour (here walking) with cognitive performance and have thus a change **yielding new conceptual understanding of the links between mind and body.**

- We thank you for appreciating the novelty and innovation of our work.

*In the experiments, participants were either standing still or walking with a constant speed, while performing a near threshold detection task. The characterization of the performance in the near threshold detection task was comprehensive. It was found that performance oscillates within the stride-cycle so that for accuracy, 14/36 participants demonstrated significant oscillations at approximately 2 cps or faster and that participant performance was clustered in phase. **However, the weakness of the study was that walking was fixed to 1.1 m/s and thus stride cycles length was always constant. As there was no variability in the cycle length that could be utilized in the analysis, it is not possible to draw firm conclusions that stride cycle would be the basis oscillations in behavioral performance.***

- We wish to clarify one point before continuing, as it can aid the reviewer's interpretation of our study and our subsequent response. While the speed of the stimulus was fixed by our motion-guide, individual stride-lengths did indeed exhibit variability within and between participants. This is because individual stride-lengths correlate (imperfectly) with height, and are determined by leg length and preferred step duration. This is the reason why we normalised our results to percentage of an individual's stride-cycle, as is standard of studies of gait – but this might have obscured this variability. We have now included summary descriptive statistics of stride-cycle variability across participants in our revised manuscript:

Lines 150-156

This analysis revealed that step- and stride-duration varied across participants (step duration $M = 0.59$ sec, $SD = 0.034$; stride $M = 1.18$, $SD = .069$). Supplementary Figure 4 visualises an example participant and the group-level stride-cycle variability. As is common in gait-based research, we next resampled all strides to 1-100% stride completion, to facilitate within- and across-participant averaging for our main effect of interest – whether the relative phase of an individual's stride-cycle would modulate visual detection performance.

- An example of this within-participant and across-participant variability is now displayed in an additional supplementary figure, reproduced below:

Supplementary Figure 4. Example stride-cycle variability. a) The top panel displays the change in head-height (detrended) for all strides of a representative participant. The histogram of all stride durations ($n=973$) is displayed in the bottom panel. b) Top panel overlays all recorded strides, all participants, with the bottom panel showing the histogram of all stride durations. c) Top panel shows the average stride-cycle per participant, prior to resampling along the x-axis to 1-100% stride completion. The distribution of average participant stride durations is displayed in the bottom panel.

- We wish to point-out that it is because of this variability in stride-lengths that we must resample all strides to 1-100% completion, as is common practice in gait-based research. It facilitates within- and across-participant averaging, and allows us to test our key hypothesis of interest: that the *phase* of a stride-cycle determines perceptual outcomes. We note that time since stride-onset is an inaccurate alternative measure, as 600 ms after stride onset may align with the swing or stance phase of separate subjects, based on their stride length.

Even more so, as the performance oscillations were not compared between walking and standing still.

- We agree it would be interesting to study oscillations while standing stationary, however it is not possible to compare based on stride-cycle in the current paradigm (as there is no trough in head-position to define step-onset when standing still). As an alternative (provided below), we have included a control

analysis analysing performance oscillations over time, in both the walking and standing data.

- We wish to reiterate that our main focus in the present study was to examine changes in performance within the stride-cycle, as previous work by other researchers has examined the average changes in performance that occur when stationary vs moving and within cycle effects on perception have never been examined.
- For this reason we deliberately included more trials when walking (than stationary), a detail which was previously hidden in our methods. We have now emphasised this detail in our main text and methods:

Lines 89-91

Below, we first compare average performance between walking and stationary conditions, before turning to our main focus, which was whether performance would modulate relative to the phase of the stride cycle.

Lines 483-487

Through pilot testing, we observed that accuracy quickly settled to approximately 75% in both conditions due to our adaptive staircase procedure, and consequently increased the trial count for the walking condition to enable detailed stride-cycle based analyses, which was our main focus of interest.

Major

*- The performance in near threshold detection task was evaluated as oscillations in the Accuracy and Reaction times. It was found that both individual level and group level data showed clear oscillations in the accuracy with the stride cycle. However, as the walking was fixed to 1.1 m/s, the cycle length of stride was fixed and there was no variance in the stride cycle length that could be utilized in the analysis. **To obtain more conclusive evidence for that stride cycle drives perceptual oscillations, it should be shown that perceptual oscillations depend on the stride cycle length.***

- Our resampling procedure – wherein performance is analysed with respect to 1-100% stride completion effectively shows that the stride-cycle length **does** determine perceptual oscillations. When people walk faster (and have a shorter stride-cycle length), the same 2-cycles per stride occurs over a shorter unit of time.
- In other words, if oscillations were not time-locked to the start and end of each stride, then resampling could shift the peak phase of performance to remove any group-level effects. A example is shown in **Review figure 2** below:

Review Figure 2: Computational example to show how perceptual oscillations are determined by stride-cycle length (expressed as a duration). **a)** The raw head position of 2 example participants is displayed, with unique stride-cycle durations. Participant 1 (broken line) has a shorter stride duration. **b)** The effect of resampling head position to 1-100% stride-cycle completion, as is performed in our manuscript. **c)** Example hypothetical data, wherein perceptual oscillations in reaction-time are determined by stride-cycle duration. Note that without resampling, these would be unique frequencies (in Hz), such that f_1 (participant 1- broken line) is faster than f_2 . **d)** Critically, resampling the oscillations in (c) per stride-cycle completion (1-100%) reveals that these oscillations are aligned, and matched in cycles per stride (cps). Thus stride-cycle duration determines the frequency of oscillations, qualitatively reproducing the group-level effects we observe. **e)** Notes an alternative case, if stride-cycle was not determined by stride-cycle duration. Here, oscillations are defined by time, and not stride-cycle duration. **f)** Displays how resampling the oscillations in (e) would smear any group-level effects, which is inconsistent with our observed group-level data.

-If the participants would walk faster or slower, would also the performance begin to oscillate in different frequency? I think it is not possible to draw firm conclusions on that performance fluctuations are caused by stride cycle, if such data is not provided.

- We have performed an additional analysis to investigate whether the idiosyncratic differences in peak frequency (e.g. 2 vs 4 cycles per stride) correlate with raw stride-cycle duration.
- Hypothetically, a strong correlation might suggest that a single oscillator (say 4 Hz), could be captured as a 2 cps or 4 cps oscillation, within individuals with a 500 ms vs 1000 ms stride-duration, respectively.
- In our data, there is no relationship between cps and raw stride-cycle duration. We have now included this exploratory analysis in our manuscript, reproducing the supplementary figure below. We note that experiments with calibrated and balanced slow and fast walking speeds are also planned in our laboratory.

Lines 248-258

We performed additional exploratory analyses to investigate whether these idiosyncratic frequencies correlated with raw stride-cycle duration. One possibility is that these oscillations are not determined by stride-cycle, but a more general oscillator such as clock-time, which when resampled over different stride-lengths could result in the variability in peak cps we observe. As a toy example, a single oscillator at 4 Hz could be captured as a 2 cps or 4 cps oscillation, within individuals with a 500 ms vs 1000 ms stride-length, respectively. In our data, we observed no relationship between the across-participant stride length and the strength of 2 cps fits, strength of 4 cps fits, nor the frequency of maximum fit strength, for any of our dependent variables (Supplementary Figure 5). Thus, it is the phase of the gait cycle that matters, not the duration of the gait. When aligned by phase, clear group effects emerge.

Supplementary Figure 5. No correlation between participant stride-cycle duration and the idiosyncratic oscillations in perceptual performance. a-c) Display scatter plots ($N=36$) of the strength of Fourier fits at 2 cps (max between 1.8 and 2.2 cps) and average participant stride-cycle duration, for accuracy, reaction time, and response count data respectively. d-f) Display the relationship between the strength of Fourier fits at 4 cps (max between 3.8 and 4.2 cps) and stride-cycle duration. g-i) Displays the correlation between the frequency (in cps) at which the maximum fit was reached and stride-cycle duration. All correlations are non-significant after correction for multiple comparisons.

- The main manipulation in the experiment was that participant spent either standing still or walked fixed to 1.1 m/s. Yet, there are no utilization of this main manipulation in the analysis of perceptual oscillation, but all analysis are about the stride length influencing performance. **As this was not explicitly stated anywhere (neither in figures, figure legends or main text), I might also be wrong.**

- Our main manipulation was to vary the timing of target onsets within the stride-cycle, and was not the comparison between standing and walking conditions.

- Previous research has already compared performance when standing still and walking (or after light-exercise). Our novel contribution is to show that performance *during walking* fluctuates.
- Our design included only 20% stationary trials, in order to increase the data necessary for our main analysis of stride-cycle modulations.

To establish that walking (not stride cycle, as it was not manipulated) is associated with perceptual oscillations, it should be compared against standing still which should show any perceptual oscillations if they are to arise from walking.

- We may misunderstand this request, but it is not possible to compare stride-cycle oscillations between standing and walking, as there are no stride onsets in standing data.
- As detailed above, we have now completed additional analyses (based on across participant stride-cycle variability), as a control and complement to the previously presented main results.
- To enable a comparison with standing data, we have now also analysed changes in performance over time, when time-locked to the start of each standing trial.
- Critically, none of the Fourier fits for perceptual oscillations in Hz based on clock-time exceed those obtained using cycles-per-stride (cf. **Supplementary Figure 7** below).

Supplementary Figure 7. An analysis of perceptual oscillations when stationary, aligned to clock-time. **a)** plots the distribution of all target onsets within each 9 second trial, averaged across participants. The oscillatory structure in target onsets is expected based on our jittered-target spacing. Critically, we note that despite this structure, the presentation of target onsets within a stride was approximately uniform (cf. **Manuscript Figure 3C**), owing to the variability of stride times across participants. **b)** Displays the average over all consecutive 1 second epochs (20 ms bins, no overlap). No oscillations were present above the 95% CI of shuffled data. **c)** displays the goodness-of-fit for all frequencies in Hz in black. Overlaid in grey is the equivalent strength of Fourier fits using the cycles-per-stride data (i.e. **Manuscript Figure 3**). Broken lines display the 95% CI for data aligned to clock-time. **d)** Displays the change in target accuracy over trial-time, while walking. **e)** displays the change in accuracy on average over 1 second epochs. **f)** Demonstrates that the strength of accuracy oscillations per stride-cycle (grey) far exceeds the strength of any oscillation per second (blue). **g,h,i)** The same data for reaction times in red. Again note that the fit-strength when aligning to cycles-per-stride far exceeds oscillations in Hz. **j,k,l)** Displays the response counts per trial time, per second, and fit strength, respectively. Although there is temporal structure owing to our target spacing, fit strength during cycles-per-stride far exceeds fit-strength in units of time, demonstrating that perceptual oscillations are time-locked to the stride-cycle.

- *The participants were to walk a **5 metre distance** at a steady speed, with the smooth duration and speed of travel set by the walking guide. How was this controlled?*

- Participants walked approximately 9.5 m during walking trials. The speed of travel was fixed by the movement of the walking guide - a 3-Dimensional object that was animated within the virtual reality environment. The display screen was aligned with the walking guide, so participants maintained the set speed to perform the task.

Is it possible that participants did not obey this instruction and walked in a different speed?

- It did happen rarely, during practice trials. Fortunately, it was always quickly noticed by whoever was conducting the experiment, as we can see the participants eye-view of the virtual environment (i.e. in-game view). As soon as participants lagged behind the moving screen, they were instructed to speed up on the next trial.

Data of the stride cycle durations at the group level should be presented if the participants were not physically bound to walk with a fixed speed e.g. with a treadmill. If there is variability in stride cycle duration, this could be utilized to test if the perceptual oscillations vary as a function stride cycle duration.

- As shown above, this data is now included in **Supplementary Figure 5**.

Minor

- The set-up is quite difficult to understand from the method section. **A schematic illustration of the setup would be helpful** to understand how participants were positioned and how they performed the experiment.

- We have now reformatted Figure 1, to demonstrate the paradigm more clearly (reproduced below). Changes include stimulus dimensions in degrees of visual angle, updated stimulus timing information, and an example trial of 3D head motion.

Figure 1. Environment and trial structure. **a)** Third-person view of the virtual environment. Participants were positioned behind a virtual grey screen displaying the target stimulus. During the trial, the screen progressed with smooth linear locomotion at a constant velocity, in line with a small walking guide (3-Dimensional animated game object). The avatar shown is for illustrative purposes only and was not present during the experiment. **b)** The visual detection task required participants to monitor a drifting circular annulus. Small target ellipses (1.7 d.v.a, 20 ms duration) appeared with a variable ITI, responses were provided via right trigger click. **c)** Example data from a single walking trial. Three-dimensional head position is recorded at 90 Hz (shown in blue). Walking produces a stereotyped sinusoidal pattern of head motion on the vertical axis (head-height, 2D projection on back-wall shown in grey). Peaks and troughs in head height correspond to the swing and stance phases of each step, respectively (see Methods). Our primary interest was whether the timing of target onset relative to stride-cycle phase would modulate task performance.

- Also, the description of the near threshold detection task was rather superficial and should be more detailed. **What is the stimulus length, timing, color and luminance and what is the inter-trial interval?**

- Stimulus length was 1.7 d.v.a, duration was 0.2 seconds.
- We did not measure luminance within the headset, but chose instead to staircase the contrast of the target from mid-grey which matched the screen background (RGB [0.4, 0.4, 0.4]), to white.
- We note that as the adaptive staircase matched hit-rate performance to approximately 75% across subjects, the subjective contrast was approximately matched between subjects. This was critical to compare within-participant fluctuations in performance over the stride-cycle.
- The average inter-trial interval was 0.92 seconds. We have now updated our Methods text below:

Lines 489-498

A maximum of eight targets were presented within each trial, with a minimum inter-trial interval of 0.8 seconds. Each target was 20 ms in duration with a titrated contrast determined by adaptive staircase (see **Staircase Procedure** below). The spacing of targets was relative to predefined anchor points (e.g. target 1 at 1.1s, target 2 at 2.3 s ...), with an additional jitter (± 0.5 seconds) to decrease the predictability of target onset within each trial (while ensuring the minimum ITI was satisfied). To further decrease the predictability of target onset times, each target had a 10% chance of being withheld from presentation, resulting in a range of actual target presentation counts per trial per participant (Min = 0, Max = 8, $M = 6.99$, $SD = 0.07$), as well as increased range of inter-trial intervals overall ($M = 0.92$ seconds, $SD = 0.10$ s).

- The fig 5 nicely illustrates that performance oscillations are clustered in phase. **Could you better explain to what phase in stride cycle, these phase distributions corresponds to?**

- We now display on Figure 5 the exact time-point that the phase-clustering is estimated from.

Reviewer #3 (Remarks to the Author):

This paper presents an exciting and to my knowledge novel result about how visual perception may be influenced by the gait cycle during natural walking. Using a VR system they are able to present a laboratory-style visual detection task during short segments of natural walking. Their results show that hit-rate, reaction time and response counts appear to follow a sinusoidal modulation over the stride cycle. To me this seems highly significant

and would be of broad interest. I have some comments on the statistical methodology which I outline below.

- We are thrilled you share our excitement and appraisal of the significance of our work, thank you.

Major comments

I have two concerns about the statistical analysis presented. The first is the lack of correction for multiple comparisons and the second is about the shuffling procedure used to generate samples from a null model.

Multiple Comparison Correction (over frequency models considered)

*The key statistical analysis of the paper quantifies how well the pattern of behavioural effects can be fit by a Fourier oscillatory model of different frequencies. This leads to the problem of multiple comparisons – a number of statistical inferences are performed, one for each frequency considered. This inflates the false positive rate of the test. When performing a single test at $p < 0.05$ the false positive rate is controlled at 0.05. When performing 50 tests (as for the 50 frequency values considered here), the overall false positive rate is now $1 - \text{binocdf}(0, 50, 0.05) = 0.92$. This means, in an experiment fitting 50 frequency models, would expect at least one frequency to provide a significant fit in 92% of experiments carried out, even if the overall null was true at every frequency (i.e. there was no oscillatory effect at all). This applies both for inferences at the group level, but particularly for the within-participant analysis. In this case one might argue they are only testing 2 specific frequencies (2 and 4 cps), but these have obviously been chosen after observing the data, and the “any” effect inference is incorrect (because the false positive rate of any effect is not 0.05). **A way to address this would be use the maximum statistics procedure with the shuffling approach they currently use (but see next point). For this, instead of taking the 95th percentile of the 1000 shuffled permutations *for each frequency separately*, they should take the maximum over frequencies, for each permutation, and then look at the 95th percentile of this maximum.***

This gives you a value of R^2 which you might expect to see in one of the 50 frequencies considered, under the null hypothesis, so this threshold controls the family-wise error rate of the cross-frequency test procedure to be 0.05 (which is required for prevalence estimation of the “any” effect, and protects against double dipping in terms of reporting prevalence of 2/4 cps).

- This is an excellent suggestion which we have implemented in our data analysis pipeline.
- Specifically, on each permutation of the data, we now retain the maximum R^2 value from all frequencies (0.1–10 cps in steps of 0.1). We now plot the 95th percentile of this maximum in our corresponding figures.
- In addition, we have updated our Methods text as below:

Lines 609-612

We compared the observed R^2 value, to the upper 95% Confidence Interval (CI) of the null distribution across all frequencies and interpreted the presence of a significant oscillation when the observed R^2 value exceeded this critical value (effectively controlling for multiple comparisons).

Shuffling Procedure

Notwithstanding the multiple comparisons across frequencies as outlined above, to determine the significance of each specific frequency the authors shuffle the data across time to create a surrogate distribution. This removes both periodic and aperiodic structure as noted in this important and relevant paper: <https://www.nature.com/articles/s41562-022-01364-0>

The situation here is slightly different to the one described in that paper (which is usually about ongoing dense sampling, rather than studying entrainment to a separate fixed cycle, here stride). But I think the key point remains. It would be better to use a shuffling procedure that preserves the temporal autocorrelation of the data. One way to do this would be rather than shuffling trial events across the stride cycle, randomly shuffle the stride-cycle onsets across the experiment and repeat the analysis.

- We are in agreement that the noted paper is an important one, two authors (MD, DA) provided invited peer-reviews on this topic. For reasons we expand on below, we believe comparing our empirical data to an aperiodic surrogate is inappropriate for our dataset, particularly when assessing oscillations in the 2 cps (or Hz) range.
- As noted, shuffling in time removes the aperiodic (e.g. $1/f$) component, and thus peaks at low frequencies of the empirical data may be artificially inflated when compared to confidence-intervals of surrogate data without aperiodic structure. While we welcome increased statistical rigour, our own assessment, and that of our peers (e.g. Re et al., 2022; Fiebelkorn, 2023) has identified that the alternative methods proposed by Brookshire suffer from unrealistically low true-positive rates. This is particularly the case when testing for oscillations at low-frequencies (as we do in the present manuscript), compared to a surrogate dataset with preserved aperiodic coefficients.
- For illustrative purposes, we reproduce Brookshire's own Figure 4b. Each cell displays the recovery rate of the surrogate method to detect significant oscillations in simulated data (1000 experiments per cell). When creating surrogate data based on an auto-regressive model, not a single oscillation is recovered at 2 Hz. Even when simulating a very large (60%) change in behavioural performance.

Review Figure 4. Reproduced Figure 4b from Brookshire (2022). Each cell displays the proportion of 10000 simulated experiments that correctly recover an oscillation when analysed with a method creating null-data with preserved aperiodic structure (AR auto-regressive fits). Note the unrealistically low true-positive rate at low-frequencies, despite very high behavioural oscillatory amplitude.

- A similar problem is reproduced in our dataset, when we use a shuffling procedure that preserves the autocorrelation of our data. (We note you also offered an alternate shuffling procedure, the advantage of our selection is that the autocorrelation of the empirical data is quantified using an auto-regressive model that captures the empirical aperiodic component).
- Specifically, we now generate surrogate datasets using the coefficients of an auto-regressive (AR) model fit to participant-level data. Our original binning and Fourier fitting analysis is then reproduced on this alternative that has preserved the aperiodic structure.
- **Review Figure 4a** below displays the results of this analysis for the empirical oscillations in Accuracy we have reported. We note that the upper-bound of the 95% CI (1000 permutations) at *each frequency* exceeds the empirical data until approximately 4 cps. The corresponding thresholds for significance at higher frequencies (>6 cps) are very low. More tellingly, the upper 95% CI of all frequencies (as proposed in the reviewer's previous comment), approaches an R^2 value of 0.8. The same results were obtained when creating permutations using an auto-regressive fit to the group level data.
- This problem arises as while the average spectrum of aperiodic trials contains no periodicity, individual spectra can contain large peaks, particularly at low-frequencies (For similar demonstrations see Re et al., 2022; examples from our data are provided below).

Review Figure 4. High statistical thresholds for significance when testing low-frequency oscillations against aperiodic surrogate data. **a)** Reproduces the strength of the group-level fit for oscillations in Accuracy at various cycles-per-stride in blue (0.1-10 cps). The black broken line displays the upper 95% CI obtained per frequency from $n=1000$ permutations testing Fourier fits on surrogate data which preserves individual participant aperiodic structure. The horizontal red broken line is the upper 95% CI using all frequencies per permutation, as requested in the previous comment. **b)** Illustrates the problem. Individually simulated aperiodic data from the using the parameters of the auto-regressive (AR) fit can progress monotonically, or have strong periodicity. This inflates the 95% CI of all R^2 values obtainable per frequency at low frequencies especially (and thus across all frequencies), **c)** displays the Fourier fits for trials in **b)**. **d)** Displays the result of averaging the spectra of surrogate trials created from the aperiodic components estimated with AR fits (cf. Re et al., 2022). Here $N=36$ fit-spectra were first averaged to create a group level Fourier spectrum, repeated $n=1000$ times. While this procedure corrects for the conservative threshold, this pipeline is a significant departure from our main analysis.

- In summary, these unrealistically high thresholds for true-positives are what detracts from the use of aperiodic surrogate datasets when testing for behavioural oscillations at lower frequencies. An alternative, proposed by Re et al., (2022), is to average the Fourier spectrum of many surrogate trials. While this is possible, it is a further departure from our analysis on empirical data. We do not average spectra, but have fit a single-group level fourier model, and tested for the prevalence of oscillations at the participant level with Bayesian prevalence estimates.
- Nevertheless, and as noted by the reviewer, our situation is quite different when compared to the main pipeline critiqued by Brookshire, which includes detrending, demeaning, windowing, zero-padding, and a multi-taper FFT on cue-evoked time-series – none of which we do in our current paper.

- As we note that your next comment includes you are “convinced that there is an oscillatory pattern in the visual behavioural variables”, we have elected to incorporate your prior recommendation for multiple-comparison correction over all frequencies, and omit an analysis based on aperiodic surrogate datasets.

I also have a concern about the overall interpretation. While, notwithstanding the above points **I am convinced that there is an oscillatory pattern in the visual behavioural variables considered**, I miss any kind of control analysis to demonstrate that these are locked to the stride cycle specifically, rather than being a less-specific oscillatory phenomenon. **For example, I would want to see some sort of control, perhaps looking at performance with respect to phase of an oscillation aligned to trial onset, would an oscillatory model fit there and would the effects be weaker than the stride alignment.**

- We thank the reviewer for this excellent suggestion, which we have now incorporated into our manuscript (Supplementary Figure 6). As predicted, stride-cycle alignment is a superior fit for all visual behavioural variables considered, compared to clock time.
- We have now included a description of this analysis in our analysis, and results in our main text. The new figure (reproduced below), demonstrates that trial-time is a poorer reference for the behavioural modulations than stride-cycle time.

Supplementary Figure 6. An analysis of perceptual oscillations when walking, aligned to clock-time. **a)** plots the distribution of all target onsets within each 9 second trial, averaged across participants. The oscillatory structure in target onsets is expected based on our jittered-target spacing. Critically, we note that despite this structure, the

presentation of target onsets within a stride was approximately uniform (cf. **Manuscript Figure 3C**), owing to the variability of stride times across participants. **b)** Displays the average over all consecutive 1 second epochs (20 ms bins, no overlap). No oscillations were present above the 95% CI of shuffled data. **c)** displays the goodness-of-fit for all frequencies in Hz in black. Overlaid in grey is the equivalent strength of Fourier fits using the cycles-per-stride data (i.e. **Manuscript Figure 3**). Broken lines display the 95% CI for data aligned to clock-time. **d)** Displays the change in target accuracy over trial-time, while walking. **e)** displays the change in accuracy on average over 1 second epochs. **f)** Demonstrates that the strength of accuracy oscillations per stride-cycle (grey) far exceeds the strength of any oscillation per second (blue). **g,h,i)** The same data for reaction times in red. Again note that the fit-strength when aligning to cycles-per-stride far exceeds oscillations in Hz. **k,l,m)** Displays the response counts per trial time, per second, and fit strength, respectively. Although there is temporal structure owing to our target spacing, fit strength during cycles-per-stride far exceeds fit-strength in units of time, demonstrating that perceptual oscillations are locked to phase of the stride-cycle.

Is stride phase a better reference for behavioural modulation than say clock-phase locked to trial onset.

- As the right-most panels in the figure above demonstrate, clock-phase locked to trial onset is not a better reference for the behavioural modulations. The oscillations are all better fit by stride-cycle alignment.

*I think related to this there could be **more explicit discussion of potential confounds** in the discussion. **For example, the authors mention that respiration can be linked to the gait cycle, but they don't explicitly note how that could be a confound here. How could future work look to determine whether this visual perception is more strongly influenced by the respiratory cycle or the gait cycle (presumably those would have to be decoupled experimentally).***

- We have now expanded our discussion of potential confounds with additional suggestions for future research. One possibility is that the across-participant variance we have captured is related to physical exertion. For example, in our follow up experiments (data collection ongoing), we have individually calibrated walking speeds, to remove this possibility.
- The respiratory rate (and cardiac cycle) can be experimentally decoupled by increasing the physical exertion of participants while maintaining the same walking speed. We are currently planning a study with collaborators using both weighted backpacks, as well as incline walking, to decouple these potential confounds.
- These details have now been incorporated into our discussion:

Lines 388-393

One intriguing possibility is that the variations in idiosyncratic peak frequency we

report are not coupled to stride-cycle duration (cf. Supplementary Figure 4) but cardiorespiratory exertion. Future studies can individually calibrate walking speeds, as well as experimentally decouple the cardio-respiratory cycle from the stride-cycle (such as while carrying a backpack, or during inclined walking), to investigate this potential confound.

*Similarly, Supplementary Figure 3 shows smooth change in gaze origin, which follows an **sinusoidal cycle over a single step**. But this shows exactly the same pattern as the visual behaviour, **so is this not a confound for that?***

- The consistent gaze shift (likely achieved via orbital rotation) may contribute to ~ 2 cps modulations, but it is unclear how they would account for the large proportion of our sample with faster cycles in behavioural modulations.
- Related to the concerns raised by Reviewer 1, walking is inherently a multi-faceted process, requiring the coordination (and synchronization) of a host of bodily systems. Many of these influences will also entrain to the rhythm of the stride-cycle (such as eye-movements, respiration, heart rate, etc), and the relative contribution of each system to perceptual behaviour is beyond the scope of the present work. We have plans for future work to replay the oscillatory trajectory of walking trials in a passive condition (using a motion simulator). This future work may quantify the relative influence of these eye movements on the novel oscillations we report.

Minor comments

*Figure 3: **why use a 0-100% scale rather than an explicit angular scale** (to make clear the circular nature of the range), either radians or degrees.*

- We use a 0-100% scale primarily because the phases of the stride-cycle are ballistic, and many individual strides contain asymmetries that are not well captured by a sinusoidal model (e.g., where one step is shorter than the other). Thus 0-100% stride cycle completion is preferable because some stride-cycles are not perfectly sinusoidal and this approach is widely used in the gait literature.

L187: the 4cps is already visible in the group mean Fig 3d, so perhaps this could be commented on.

- The 4cps oscillation is no longer as prominent in group data after our revised threshold for saccade and blink detection, as new numbers of trials are included per participant compared to the previous manuscript (cf. Reviewer 1).

There doesn't seem to be much difference between the two strides of a cycle (I guess left and right), so perhaps they would have more statistical power (noting the extra controls

needed above) looking at phase over a single leg movement, ignoring whether it is left or right. Difference between left and right stride could be tested explicitly. Single-step analysis is reference on line 503, but I don't see the results featured very prominently (not sure where these single-step analyses are featured).

- We initially began with a single-step analysis, but noted that the resampling procedure can occasionally introduce small edge-artefacts, and that sinusoidal fits were poorer (owing to fewer cycles).
- We thus settled on a stride-based analysis, to ensure the changes in performance we measure at footfall are reproduced in the centre of a resampled epoch (i.e. away from edges), and to visualise the sinusoidal fits.

In discussion, could maybe talk about more about neural oscillations. Could the heel strike or some other discrete part of the stride cycle be causing a phase-reset, would these fit into known behavioural oscillations.

- This is an excellent idea we are pursuing with mobile-EEG in our laboratory. We have provided citations and a link to this literature in our discussion.

Lines 355-358

Similarly, reaching behaviour has been shown to reset the phase of endogenous oscillations (Tomassini et al. 2017; Tomassini et al. 2020; Benedetto et al. 2020) and the ballistic movements generating locomotion may reset or entrain neural oscillations relevant to behavioural performance (Schroeder et al., 2010).

REVIEWERS' COMMENTS

Reviewer #1 (Remarks to the Author):

All my point have been adequately addressed and I have no further concerns.

Reviewer #2 (Remarks to the Author):

The authors have done a great job in revising the manuscript "Walking entrains unique oscillations in performance on a visual detection task" by Matthewson et al.,. The new control analyses required in the revision helped to establish more rigorously that reaction times in near sensory detection task show modulation with the stride cycle modulating at approximately 2 cycles-per-stride (~ 2 Hz). I still have some minor suggestions on how further to improve the manuscript and establish the link between walking and perception at the individual level.

Minor

- The authors have added a supplementary Supplementary Figure 4 showing examples of stride-cycle variability across individuals. In my opinion, this is quite central in establishing that i) there is variability in stride cycle duration, and that ii) this variability impacts performance together with new analysis aligned to clock-time. I would suggest adding the bottom panel to Figure 1.
- The authors argue that When people walk faster (and have a shorter stride-cycle length), the same 2- cycles per stride occurs over a shorter unit of time and if oscillations were not time-locked to the start and end of each stride, then resampling could shift the peak phase of performance to remove any group-level effects. In the rebuttal it is explained that at the individual level perceptual cycle length vary by the length of the stride cycle length, but supplementary figure 6 does not have any plot showing individual variability in perceptual cycle vs. stride cycle lengths, which should be the case if these covary. Note that this is complementary evidence to that perception is not based on trial time which is now the main message of supplementary figure 6 and could be shown as a scatter plot in figure 4.
- line 208. In addition to the group-level test of oscillations in performance, we tested the 209 population prevalence of oscillations within our sample. Be specific here and elsewhere whether you referring to perceptual or stride cycle oscillations.

Reviewer #3 (Remarks to the Author):

I think the authors have produced an excellent and comprehensive response that addresses my concerns and those of the other reviewers.

I have only one small comment on what I hope to be a typo. For the maximum statistics approach to multiple comparison correction it is crucial that for each permutation the maximum over all 50 frequencies is taken, and the 95th percentile of these max-over-frequencies is obtained. The new text added in the revision doesn't quite describe this:

"We compared the observed R2 value, to the upper 95% Confidence Interval (CI) of the null distribution across all frequencies". If it is the 95th percentile of all frequencies over all permutations this is not correct. Taking the max over frequencies within each permutation is crucial.

In the text of the response this is described correctly so just want to clarify and correct this is in the manuscript if necessary.

"Specifically, on each permutation of the data, we now retain the maximum R2 value from all frequencies (0.1–10 cps in steps of 0.1). We now plot the 95th percentile of this maximum in our corresponding figures."

AUTHOR RESPONSE

We thank the editorial team and expert reviewers for their continued time and suggestions for improving our manuscript. Below, we provide a point-by-point reply. Reviewer's comments are *italicised* and our emphasis is in ***bold italics***. Key changes to the revised manuscript are included with line numbers in blue text.

REVIEWERS' COMMENTS

Reviewer #1 (Remarks to the Author):

All my point have been adequately addressed and I have no further concerns.

- Thank you kindly for your contributions to our manuscript.

Reviewer #2 (Remarks to the Author):

*The authors have done a great job in revising the manuscript "Walking entrains unique oscillations in performance on a visual detection task" by Matthewson et al.,. The new control analyses required in the revision helped to establish more rigorously that reaction times in near sensory detection task show modulation with the stride cycle modulating at approximately 2 cycles-per-stride (~2 Hz). ***I still have some minor suggestions*** on how further to improve the manuscript and establish the link between walking and perception at the individual level.*

Minor

• *The authors have added a supplementary Supplementary Figure 4 showing examples of stride-cycle variability across individuals. In my opinion, this is quite central in establishing that i) there is variability in stride cycle duration, and that ii) this variability impacts performance together with new analysis aligned to clock-time. ***I would suggest adding the bottom panel to Figure 1.****

- We have now included the example of across-participant stride-cycle variability (supp Fig 4c, f), in our revised Figure 1. Reproduced below:

Figure 1. Environment and trial structure. **a)** Third-person view of the virtual environment. Participants were positioned behind a virtual grey screen displaying the target stimulus. During the trial, the screen progressed with smooth linear locomotion at a constant velocity, in line with a small walking guide (3-Dimensional animated game object). The avatar shown is for illustrative purposes only and was not present during the experiment. **b)** The visual detection task required participants to monitor a drifting circular annulus. Small target ellipses (1.7 d.v.a, 20 ms duration) appeared with a variable ITI, responses were provided via right trigger click. **c)** Example data from a single walking trial. Three-dimensional head position is recorded at 90 Hz (shown in magenta). Walking produces a stereotyped sinusoidal pattern of head motion on the vertical axis (head-height, 2D projection on back-wall shown in grey). Peaks and troughs in head height correspond to the swing and stance phases of each step, respectively (see Methods). **d)** Average detrended head-height for each participant over their respective stride-cycle. **e)** Distribution of average stride-cycle duration across participants. Our primary interest was whether the timing of target onset relative to stride-cycle phase would modulate task performance.

• The authors argue that When people walk faster (and have a shorter stride-cycle length), the same 2- cycles per stride occurs over a shorter unit of time and if oscillations were not time-locked to the start and end of each stride, then resampling could shift the peak phase of performance to remove any group-level effects. **In the rebuttal it is explained that at the individual level perceptual cycle length vary by the length of the stride cycle length, but supplementary figure 6 does not have any plot showing individual variability in perceptual cycle vs. stride cycle lengths, which should be the case if these covary. Note that this is complementary evidence to that perception is not based on trial time which is now the main message of supplementary figure 6 and could be shown as a scatter plot in figure 4.**

- We appreciate the suggestion and have now calculated the participant-level perceptual oscillations in Hz (before resampling over the stride-cycle to 1-100% stride completion). The frequency of the best fit (in Hz) can be plotted against an individual's stride-cycle duration, to test our argument that when people walk faster (and have a shorter stride-cycle duration), then the frequency of their oscillations must increase to fit in a shorter unit of time. This would result in a negative correlation, where faster oscillations in Hz occur at shorter stride-cycle durations.
- As in our original cycles-per-stride analysis, perceptual oscillations tend to cluster in Hz, albeit over a larger range owing to the variability in stride-cycle durations. Importantly, focusing on the correlations within these clusters demonstrates the expected negative correlation: at shorter stride-cycle durations, perceptual oscillations are faster.
- One hesitation to including this analysis in our manuscript is that the correlations are based on small numbers of participants within each cluster. Another is that Figure 4 (prevalence statistics) is already very busy. As a result we have elected to omit this complementary analysis from Figure 4, but include it as Supplementary Figure 8, reproduced below:

Supplementary Figure 8. Relationship between perceptual oscillations in Hz and stride-cycle durations. In all panels the y-axis displays the frequency (in Hz) of the best fitting Fourier model to performance oscillations, when fitting based on stride-cycle duration (in seconds). The x-axis shows the stride-cycle duration in seconds. **a)** displays the scatter plot and non-significant correlation ($N=36$) for accuracy data. Note however the clear grouping of data for oscillations at approximately 2 and 4 Hz (shaded grey regions). This grouping suggests the group-level analysis may not capture trends within each cluster (aka Simpson's paradox). Focusing instead on the correlations within these clusters **b)** demonstrates a non-significant negative correlation between stride-duration and oscillations at approximately 2 Hz, and **c)** a significant negative correlation at approximately 4 Hz, as predicted. **d-f)** Show the same analyses on the data for oscillations in reaction times, and **g-i)** for response likelihood.

• line 208. In addition to the group-level test of oscillations in performance, we tested the 209 population prevalence of oscillations within our sample. Be specific here and elsewhere whether you referring to perceptual or stride cycle oscillations.

- We have now made the type of oscillations explicit:

Line 208: "In addition to the group-level test of oscillations in performance based on the stride-cycle, we tested the population prevalence of stride-cycle oscillations within our sample."

Reviewer #3 (Remarks to the Author):

I think the authors have produced an excellent and comprehensive response that addresses my concerns and those of the other reviewers.

I have only one small comment on what I hope to be a typo. For the maximum statistics approach to multiple comparison correction it is crucial that for each permutation the maximum over all 50 frequencies is taken, and the 95th percentile of these max-over-frequencies is obtained. The new text added in the revision doesn't quite describe this: "We compared the observed R2 value, to the upper 95% Confidence Interval (CI) of the null distribution across all frequencies". If it is the 95th percentile of all frequencies over all permutations this is not correct. Taking the max over frequencies within each permutation is crucial.

In the text of the response this is described correctly so just want to clarify and correct this is in the manuscript if necessary.

"Specifically, on each permutation of the data, we now retain the maximum R2 value from all frequencies (0.1–10 cps in steps of 0.1). We now plot the 95th percentile of this maximum in our corresponding figures."

- Thank you for the suggestions for improvement, this was indeed a typo. We have now updated the text in our manuscript to match that of our previous response:

Line 605-

We compared the observed R^2 value to a null distribution created by retaining the maximum

R^2 value across all frequencies per permutation, and interpreted the presence of a significant oscillation when the observed R^2 value exceeded the upper 95% Confidence Interval (CI) of this null distribution (effectively controlling for multiple comparisons).